# Factors affecting job performance of Sri Lankan IT professionals working from home

**Nilesh Jayanandana** [1], **Ruwan Jayathilaka** [2]*

1 SLIIT Business School, Sri Lanka Institute of Information Technology, Malabe, Sri Lanka, 2 Department of Information Management, SLIIT Business School, Sri Lanka Institute of Information Technology, Malabe, Sri Lanka

* ruwan.j@sliit.lk

**Data Availability Statement:** All relevant data are within the paper and its supporting information files.

## Abstract

This study investigated the influence of the physical work environment, work life balance, work flexibility, and effective communication on the job performance of IT professionals in Sri Lanka's IT industry who work from home (WFH). A standard questionnaire was used to collect data from 293 IT specialists in 50 different IT organizations in Sri Lanka, and a step-wise probit model was employed for data analysis. According to the findings, both the physical work environment and work life balance had a significantly positive effect on job performance. A one-unit increase in the physical work environment and work life balance increased the likelihood of high job performance by 0.21% and 0.19%, respectively. In contrast, work flexibility had a negative effect on job performance, with an increase of one unit resulting in a 0.18% decrease in the likelihood of high job performance. The positive impact of effective communication on job performance was less significant. The study emphasises the significance of providing a conducive work environment and promoting work life balance to enhance the job performance of IT professionals in Sri Lanka's IT industry who WFH.

## Introduction

The share of workers working from home (WFH) is rapidly increasing indicating that WFH has a significant presence in the cooperate sector, surpassing the traditional office-based working paradigm. This transformation was primarily driven by the necessity for businesses to maintain their operations, as WFH became the only viable option during the extended lockdowns due to social distancing rules. These regulations were implemented worldwide to contain the deadly spread of the COVID-19 pandemic, which later became the new normal in the post-2020 world. WFH is a subset of the larger issue of remote working, which has been around for decades, derived from the concept of telework and continues to grow [1]. Consequently, the concept of WFH or telework, is not novel in the corporate sector globally. However, this concept is relatively new to the work environment norms in Sri Lanka, which traditionally required employees to be physically present at the workplace.

Due to technology improvements, a growing number of firms are embracing the WFH practice. In addition, with the sudden pandemic outbreak, opportunities emerged for the working population at large to familiarise and adapt to telecommuting. Employee performance

**Funding:** The authors received no specific funding for this work.

**Competing interests:** The authors have declared that no competing interests exist.

is defined as the extent to which employees effectively carry out their assigned job responsibilities in accordance with the company's requirements. Another research exploring factors affecting employee performance in 2018 defined employee performance as the number of outcomes achieved by an individual employee for each job function during a given period [2]. A similar study done in 2020 proved that employee performance massively influence the overall success of a business [3]. Given recent changes in workplace norms and to minimise work disruptions, it is expected that WFH would shift from a voluntary to a mandatory necessity, compelling people to work full-time in their homes. In another recent study, it was determined that WFH full-time has both benefits and drawbacks [4]. One research confirmed that employee morale is enhanced by a range of factors, including 'absence'- in terms of face-to-face meetings, colleague diversions, close supervision, lengthy commutes, and family concerns when at home [5]. Similarly, another research suggests that mental health professionals are concerned about the escalation of mental health problems resulting in physical health difficulties [6].

Recent studies suggest that shifting from an office-based to a remote work paradigm may have a detrimental influence on an employee's interaction with peers, work life balance, work habits, and work performance, all of which are negatively affected to a significant extent. Recent research studies conducted by the Universities of Birmingham and Kent aimed at workers in the United Kingdom (UK). Here, most workers in the UK preferred flexible working hours but did not want to WFH regularly [7, 8]. However, the results have aroused concerns about the potential negative impacts of flexible working hours on employee productivity and performance.

The COVID-19 pandemic has significantly altered how organisations operate, with remote work gaining popularity. Furthermore, WFH policies have been implemented in worldwide to ensure business continuity and employee safety. Southeast Asian nation like Sri Lanka is not an exception to this trend. The country's thriving information and communications technology (ICT) sector heavily depends on remote work practices. However, little attention has been paid to how WFH affects the performance of IT professionals in Sri Lanka. Despite the abundant research on remote work, a focus on the performance of IT professionals during WFH is necessary, particularly in Sri Lanka. Filling this void, the present study focuses on the effects of effective communication, flexible work schedules, work life balance, and physical work environment on the performance of employees during WFH. This study will be guided by the following research questions: How does effective communication affect employee performance during WFH? How does a flexible work schedule impact WFH employee performance? How does work life balance impact the performance of employees during WFH? and how does the physical work environment influence WFH employee performance? By addressing these research questions, this study will contribute to gaining a better understanding of the factors influencing the performance of IT professionals during WFH. Thereby, the study assists organisations and policymakers in Sri Lanka in strengthening their policy management and practices regarding remote work. It is worth noting that in 2023, organizations have started to shift back to office work policies, and WFH is not encouraged as much as it used to be. This study aims to provide better insight into that decision and explore what factors have affected work-from-home for IT professionals in Sri Lanka. This study also references research work by multiple scholars over a 22-year timespan from 1998 to 2020 [4, 9–13].

In focusing on the Sri Lankan context, it is essential to highlight the unique circumstances and emerging status of the country in the realm of information technology. Sri Lanka, as a developing nation, presents a dynamic landscape where the IT sector is swiftly evolving. It is pertinent to recognize that Sri Lanka's IT literacy and workforce, in comparison to other developing countries, are factors that have remained unexplored in previous studies conducted elsewhere. For instance, exploring how Sri Lanka ranks in terms of IT literacy and its burgeoning

IT sector compared to other developing countries can shed light on the distinctive features of the Sri Lankan workforce. According to the World Bank, Sri Lanka's IT sector has grown by 20% annually in recent years, outpacing the global average of 10%. The country's IT literacy rate is 60%, which is higher than other developing countries in the region, such as India (40%) and Bangladesh (25%) [14]. These factors, in conjunction with the global shift towards WFH, underscore the need for a focused examination of the performance of IT professionals in Sri Lanka during remote work scenarios.

The theoretical framework for the study is based on the job demands-resources (JD-R) model, which suggests that job demands can lead to job strain and negative outcomes, while job resources can lead to job engagement and positive outcomes [15]. In this study, WFH is a job demand, whereas effective communication, work flexibility, work life balance, and physical work environment are job resources. The study hypothesises that these resources can mitigate the adverse effects of WFH and boost job performance. Specifically, effective communication can reduce isolation and improve collaboration, physical work environment can provide a comfortable and conducive workspace, work flexibility can provide autonomy and control, and work life balance can decrease work-family conflict and improve overall well-being. The JD-R model provides a theoretical lens through which the complex interaction between job demands and job resources, as well as how these can impact job performance in the context of WFH for IT professionals can be comprehended in Sri Lanka's IT industry.

The next sections of the article are organised as follows: the first part analyses the existing literature, followed by the data used and the technique used in this investigation, then the findings and discussion, and lastly, the general conclusion of the study with some policy recommendations.

## Literature review

This section encompasses literary contributions by numerous academics worldwide. The literature evaluation was based on 158 articles identified from reputed sources and were later categorised for a better understanding of the underlying concept and variables considered for the framework. Fig 1 depicts the identification, retention, and discarding of research during the various phases of the literature review. The search results yielded five major categories: work life balance, effective communication, working from home and performance, work flexibility, and the physical work environment, as discussed above.

A total of 158 online publications with full texts written in the English language and peer reviewed were identified from the period between January 1985 and March 2023. The abstracts were then reviewed and discarded if: (i) these were irrelevant to the topic (n = 21), (ii) these lacked sufficient information (n = 15), or (iii) these were duplicates of previously published articles (n = 35). This procedure led to the exclusion of 35 publications. Data were extracted from the remaining 87 publications and categorised into five subject areas: work life balance (n = 15), work flexibility (n = 16), effective communication (n = 16), physical work environment (n = 18), and performance (n = 22). To conclude the methodology, 87 empirical case studies were selected within the aforementioned five areas, and their key characteristics are summarised below.

The theoretical underpinning of this study is grounded in the Job Demands-Resources (JD-R) model, an influential framework in the field of organizational psychology and work-related research. Developed to examine the interplay between workplace elements and employee well-being, the JD-R model posits a fundamental distinction between two categories of factors: job demands and job resources [15]. Job demands encompass the aspects of work that require sustained physical or psychological effort and are associated with the potential for

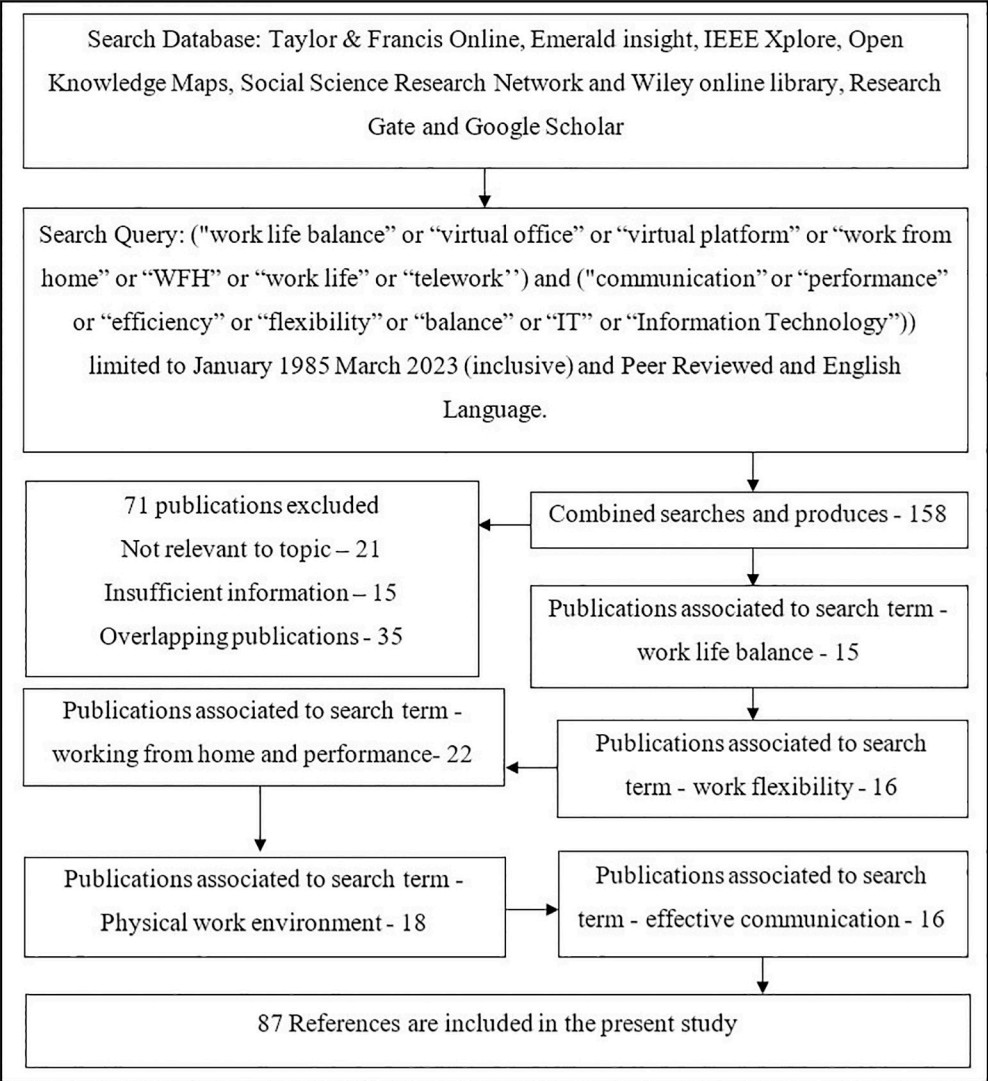

**Fig 1. Literature series flow diagram.** Source: Authors' illustrations.

strain and negative outcomes. In the context of this study, the act of WFH serves as a pertinent job demand, given its potential to introduce challenges and stressors into the professional lives of IT professionals in Sri Lanka. On the other side of this theoretical spectrum are job resources, which are the elements of work that facilitate the achievement of work goals, reduce job strain, and promote positive outcomes. In this study, we identify four critical job resources that play a pivotal role in enhancing the job performance of IT professionals during WFH: effective communication, work flexibility, work-life balance, and the physical work environment. Effective communication fosters collaboration and alleviates the isolation often associated with remote work. Work flexibility empowers employees with autonomy and control over their work schedules and conditions. Work-life balance addresses the delicate equilibrium between personal and professional life, a crucial determinant of overall well-being. The physical work environment, even in the context of remote work, provides a comfortable and conducive workspace for IT professionals. The JD-R model serves as an invaluable theoretical lens through which the complex interplay between job demands and job resources is examined. It

offers a comprehensive perspective on how these factors can influence job performance in the unique setting of WFH for IT professionals within Sri Lanka's IT industry. Through this framework, this study seeks to unravel the intricate dynamics of remote work and offer insights that can guide organizations and policymakers in enhancing work conditions and performance for IT professionals in an evolving work landscape [15].

## Working from home and employee work performance

Several literature studies stress that WFH has both positive and negative impacts on employees. In the wake of a study done in 1998, whether telecommuters perform at a low or high level was identified [9]. Here, the data demonstrated that attitudes and ideas of telecommuters regarding the quality of their relationships with their bosses and family members were highly related to their performance levels. Researchers believe that home-based telework is on the verge of take-off and is strongly associated with the social expectations of those who do the WFH. The primary factors include improved work life balance, a less stressful work environment and a reduction in fatigue caused by the lesser need to travel, similar to the findings of other research in 1999–2009 [10–12]. Further, the degree of physical isolation has adversely affected virtual employees' perceptions of their perceived respect and belongingness in the firm.

Employees who WFH are competent to integrate their personal and professional lives effectively and sustainably. The WFH practice isn't universally supported by employers, nor universally desired by employees. A research published in the Scandinavian Journal of Work, Environment & Health examined the link between work–home interference and tiredness and WFH mismatch [16]. The desire of employees for flexibility in their work schedules and the ability to WFH has led these researchers to believe that a company-specific policy for WFH may be an effective strategy to achieve a healthy work life balance for employees. In a survey of small and medium-sized enterprises (SMEs), A research carried out in 2020 analysing Greek teleworkers revealed a twofold lack of appeal for home-based teleworking for employers, which did not materialise in sufficient productivity improvements to offset the expenses of teleworking [17]. The reason was that teleworking did not result in improved work life balance for workers, which was necessary for improved performance and productivity. Contrary to the popular belief, a research analysing home-based telework in France suggested that the corporate community seemingly has not embraced the future of home-based teleworking as expected [18].

Flexibility in the workplace is traded in return for positive views toward the job, such as dedication [19–21], motivation [22] and more. These attitudes may lead to favourable outcomes, such as individual competency, proactivity, and flexibility to changes in work. Employee morale and productivity will remain high if management delegates sufficient authority or has faith in the people they employ. People are more open to change when they have a sense of control over their work results, as affirmed by multiple studies on job autonomy [23–25]. In addition, a few studies indicate that increasing subordinates' faith and trust in their superiors encourages a variety of other behaviours and productivity [26, 27]. According to studies done in 2001 and 2021, it is critical for employees who telework to have a sense of support from the company, since this encourages productive behaviour. Accordingly, the level of autonomy, trust, and support teleworkers get from their supervisors may impact their performance [28, 29].

A study conducted by Stanford University revealed that call centre workers who worked from home were 13% more productive compared to those who worked in the office [30]. This increase in productivity was attributed to several factors, such as reduced distractions, a more

comfortable work environment, and the flexibility to work during their most productive hours. However, it's worth noting that other studies have shown conflicting results regarding the impact of working from home (WFH) on productivity. For instance, a study carried out by the University of Oxford found that employees who worked remotely were more prone to taking breaks and experiencing a lack of focus [31]. This was attributed to the absence of social interaction and the temptation to engage in multitasking. In a meta-analysis of 130 studies examining WFH and productivity, it was observed that there exists a small yet significant positive relationship between WFH and productivity [32]. The researchers concluded that WFH can lead to marginal yet favourable improvements in productivity.

Various authors believe that telework requires supervisors who trust their workers and abandon control-based management in favour of a more trust-based approach. One such study examined whether the supervisors' control on teleworkers moderates the relationship between two of the three variables that measure worker individual performance [33]. Several significant business variables, including worker performance and productivity, pay, absenteeism, turnover, and overall company success have been linked to telecommuting [13]. Employee performance is defined as how successfully the workers carry out their assigned job obligations aligning with the company's standards. A company's total success is significantly influenced by the level of employee performance, defined as the number of outcomes achieved for each job role within a specified period. The main worry related to the WFH practice is the quality of job performance. Individual elements, such as knowledge, career path and competences, work style and vision, as well as natural variables such as values and philosophy, are one of the three fundamental factors determining job performance. A study done in 2021 summarises this finding by stating that people are either content and productive or disinterested and incapable of doing anything [34].

In a study done in 2017 with a sample size of 235 data entry workers in India, the authors explored how working from home affected worker productivity. They found that workers who were randomly assigned to work from home were 18% less productive than those who worked in the office. This difference was mostly due to the fact that workers who worked from home learned more slowly. The authors also found that workers who preferred to work from home were more productive than those who preferred to work in the office. However, this difference was not enough to offset the negative effects of working from home. Overall, the paper finds that working from home can have a negative impact on worker productivity. However, the authors also noted that there are some benefits to working from home, such as improved work-life balance and reduced commuting time [35]. In contrast, A study by the Sri Lanka Association of Software and Services Companies (SLASSCOM) found that Sri Lankan IT professionals performed well during remote work scenarios. The study found that 90% of IT professionals were able to maintain or improve their productivity while working remotely. Additionally, 85% of IT professionals reported that they were satisfied with their work-life balance while working remotely. Overall, the Sri Lankan IT sector is a dynamic and growing sector with a young and adaptable workforce [36]. IT professionals in Sri Lanka have performed well during remote work scenarios, demonstrating the country's potential to become a major player in the global IT industry.

## Working from home and effective communication

Free time is critical for employee rejuvenation and well-being, but so is psychological separation from work. A study investigating WFH and psychological detachment from work examined the demand for work-home division, the perception of segmentation norms, and the usage of communication technologies at home, as precursors of psychological detachment

[37]. Psychological detachment was positively connected with segmentation preference and norm. To feel informed, important, and involved, teleworkers need to feel supported and trusted. A study conducted in 2015 demonstrated that the key three factors associated with teleworking fall into three categories: support, communication, and trust [38]. A study exploring the mental and physical effects of WFH disclosed that coordination would be challenging and even infeasible in a physically split work team, especially when interdependent tasks are to be completed [39]. Early studies indicate that it is vital to maintain verbal and written communication in a remote working environment to ensure employee engagement and happiness [40]. Verbal communication aids in the development and maintenance of connections and engagement among remote working individuals, allowing employees to feel more involved and a part of a larger team. Formal communication enables one to address work-related concerns promptly. Social isolation is a frequently reported drawback of telecommuting. This is supported by the research findings of the study exploring the emotional impact of teleworkers engaged with computer mediated communication [41], among other studies. Here, these scholars highlighted the significance of personal experience with face-to-face contact, which aligns with natural human instincts and biology. In summary, interpersonal communication with a personal touch has progressed at a slower rate compared to technical advances in communication [42].

Research conducted in 2021 revealed that 70% of employers identified communication as a significant challenge for their remote workers [43]. Furthermore, a study conducted by Stanford University indicated that remote workers with strong communication skills were more likely to report job satisfaction and be recognized as high performers [44]. The quality of communication technology employed can significantly influence the effectiveness of inter-employee communication. For instance, a study published in the Journal of Organizational Behaviour [45] found that employees who utilised high-quality video conferencing technology were more likely to experience a sense of connection with their colleagues.

In summary, effective communication is essential for home-based workers, as it helps combat feelings of social isolation and disengagement. Clear communication regarding work expectations, job responsibilities, goals, objectives, and deadlines increases job satisfaction, loyalty, and productivity. Virtual communication is rife with miscommunication, which can result in decreased productivity during WFH. Consequently, it is hypothesised that effective communication has a positive effect on employee performance in a home-based office.

## Working from home and flexible work

Flexibility is the degree to which a person may exercise discretion and autonomy [27]. Work flexibility, or flexible working, can be defined as the degree to which an employee has discretion over when and where they work [46]. Flexible work can be classified into four types; Flex time, a shorter workweek, telecommuting (flex location), and part-time job arrangements [47]. However, employees who WFH have the freedom to choose their own work schedules and work environments, resulting in greater job flexibility [48].

Recent research illuminates the diverse dimensions of work flexibility and its implications for employees. In a study conducted in 2021, secondary school teachers in Ekiti State displayed a modest level of work-life balance, work flexibility, and job efficiency, underscoring the interconnectedness of work-related stress, workload, and job performance. Work-flexibility, in this context, significantly influenced work life balance, which in turn, influenced the teachers' job performance. Furthermore, insights gained from investigations into remote work experiences during the initial wave of the COVID-19 pandemic in Portugal underscored the impact of the work environment and flexible work culture on telework happiness [49]. Comprehensive

research is required to further understand telework satisfaction and its effects on physical and mental health, facilitating the development of effective strategies for enhancing employee well-being and cultivating a conducive teleworking environment. As the scope of telework and mobile work arrangements continues to expand due to advances in digitalization, it reshapes working conditions and job quality. A study published in the International Journal of Environmental Research and Public Health delved into the impact of various telework forms on job quality. The research revealed that gender, the nature of telework, flexible hours in telework and the intensity of information and communication technology (ICT) usage significantly shape working conditions and job quality [50]. Notably, home-based teleworkers who regularly work from home reported the highest job quality, underscoring the benefits of work flexibility. However, this achievement sometimes comes at the cost of lower skill development, decision-making autonomy, pay, and career opportunities. This research aims to investigate current data on job quality and work flexibility, with a particular focus on gender as a critical aspect of analysis. Regarding employee performance, a flexible work arrangement was found to have no direct impact.

As the demands of both professional and personal life grow, so does the need for effective time management and work overload. An individual's ability to function well at work is adversely affected by factors such as job overload and poor time management. In a study published in the Dynamic Relationships Management Journal [51], researchers looked at how time management affects job performance and the link between work overload and flexible working hours which lead to work life balance. The findings revealed that effective time management reduces the negative effects of work overload and poor job performance when combined. It means workplace productivity and work–life balance suffers because of overwork. Considering these results, it is essential for both people (employees) and companies (employers) to pay greater attention to time management to promote work–life balance and productivity.

In conclusion, flexible work arrangements, such as WFH or working outside of normal business hours, are associated with a positive work attitude and high job performance. These arrangements assist workers in maintaining a healthy work life balance, thereby enhancing job satisfaction and productivity. There is a correlation between flexible work arrangements and job performance, according to previous studies. The purpose of these arrangements is to assist employees in striking a balance between their personal and professional lives (i.e., work life balance), which can lead to increased productivity. Therefore, it is hypothesised that flexible work arrangements improve employee performance in a home-based office.

## Working from home and work life balance

Due to the significant relationship between home-based telework and workers' expectations, the researchers anticipate that home-based telework will see a rapid "take-off." The primary factors include improved work life balance, a less stressful work environment, and a drop in fatigue caused by travel, assert [52, 53]. Work life balance and work stress were examined in research in 2021 to fill the gaps caused by the COVID-19 influence on WFH. As explained above in this study, WFH, striking a work–life balance and dealing with work stress, directly and indirectly impacts happiness at work. As a new pace of work, WFH can potentially keep Indonesian employees happy in their existing work environment [27]. Moreover, researchers note that WFH in a collectivist environment has the potential to be a good indicator for the company's performance.

A study which explored the benefits of virtual labour, specifically for women in the Canadian area disclosed that in contrast to most findings, women's work time is further restricted due to telework; this makes them tend to their family's needs such as attending to the needs of

their kids during work hours [54]. Probably, men's participation in telework will play a crucial role in restoring gender relations in future era, where even men will have to attend the needs of their kids during work hours. Another component that contributes to a good work life balance is favourable social life. In 2021, similar research explored the link between work life balance and employee productivity in Sri Lanka. Here, employee-related features of family responsibilities have shown a substantial impact on the performance of private and public sector employees. Employee happiness, goal expectations, and employee contentment have had a positive and significant linear relationship with childcare concerns, dependent care, and employees who could spend time with their families [55].

Work life balance is emphasised in the second of two articles about Happy Computers, a UK computer training firm, published in 2008 [56, 57]. It describes the development of the company's work life balance policy and examines some of its outcomes. The study highlights that finding flexibility in the workplace contributes to employee motivation. At Happy Computers, the annual employee turnover is 10% compared to the industry average of 17%, indicating that workplace flexibility has fostered a greater sense of individual freedom and empowerment among employees.

Research suggests that work life balance practices have an inverse relationship with business effectiveness. Organizations that invest in work life balance strategies are more likely to observe positive impacts on employee happiness and performance outcome [58, 59]. Failure to effectively manage the work life balance situation can result in decreased employee productivity and performance. Telework enables employees to spend more time with their families, promoting a healthy work life balance for all involved [52]. While work stress can affect performance in a traditional office environment, the proximity to family members can also influence an individual's stress levels [60]. Several studies have shown that work life stress does not always have a detrimental effect on workplace performance and mood [1].

According to a study conducted in 2008, [61], interruptions are common, and working long shifts or hours negatively impacts a worker's ability to maintain a good work life balance. Many individuals do not have an ideal family life, and despite the home being seen as a safe haven, it can also be a source of tension and anxiety for them [62]. For instance, when the boundaries between work and family life are blurred, workers are more susceptible to frequent disruptions caused by family issues, making it challenging for them to concentrate on their work and fulfil their job requirements. Unless work life balance discussions incorporate a genuine understanding of family life, acknowledging its inherent tensions, they will remain disconnected from the reality of most people's lives.

As explained previously, WFH and flexible hours have become more critical considering the COVID-19 pandemic. Moreover, these work arrangements are required to comply with public health requirements, while these can be continued if no other solutions are available. Managing in a workplace where team members have dissimilar core working hours may be challenging. The dynamics of a team might be further strained if employees assume that their demands or preferences are treated unfairly by management or if the existing work policies are not revised to accommodate a WFH work setting. As the pandemic subsides, it's possible that some workers may be unable (due to the nature of work, job requirements etc) or reluctant to return to working entirely on-site. In this new environment, leaders will have to examine ethical problems, work out on possible options to address issues to achieve company objectives. When aiming to strike a balance between the requirements of the individual and the group, the concepts of equity, diversity, and inclusion will be crucial. Retaining top talent will need a supportive work environment and welcoming culture.

In summary behind the idea of "work life balance" is that an individual's personal and professional lives should complement one another. It has been discovered that remote work has

both positive and negative effects on work life balance, which can affect worker performance and productivity. According to prior research, a lack of assistance in balancing professional and personal responsibilities can increase stress levels and result in subpar performance. Therefore, it is hypothesised that a healthy work life balance has a positive effect on the performance of employees who WFH.

## Working from home and physical work environment

When looking at the many elements that influence WFH, it was discovered that decreased interpersonal interactions, decreased management support and trust, and the appropriateness of the work atmosphere at home-office were highly important [63]. Workers who WFH can personalise their environment to meet their own specifications, requirements and preferences, which can improve job performance [64]. A study investigating the effect of WFH on job performance during the COVID-19 crisis in Indonesia stated that WFH allows workers to create a customised work environment tailored to their preferences and way of life [65]. Employee performance in WFH scenarios is positively influenced by a physically acceptable work environment, which has a substantial and favourable effect on performance. Having a well-maintained working environment with appropriate space, a peaceful ambience, good lighting, and improved working equipment enable employees achieve even greater levels of productivity [2, 15, 66, 67]. Additional studies indicated that workplace factors and operational commitment had an impact on job performance, both directly and indirectly when WFH, which support earlier results [50, 68, 69].

The Journal of Occupational and Environmental Medicine published a cross-sectional study on the performance of home-based workers in relation to their actual work environment in 2021. The study was conducted in Japan on workers who spent at least one day per month working from home. When working with WFH, the physical work space was a source of exposure. The Work Functioning Impairment Scale was used to assess the existence of work-related impairment. A "No" answer to suggested surroundings was shown to be a significant predictor of work functional impairment. Answering "No" to the question and "Is there enough light to perform my job?" were related to the greatest odds ratio of work functional impairment [29].

A paper published in 2020 disclosed that teachers' self-motivation in a remote work environment is influenced by various factors. These include their perceptions of the setting's ability to meet their psychological requirements for autonomy, competence, and interpersonal connection. To assist team leaders in being more responsive to the needs of their subordinates, the study have provided a list of practical tips [70]. Working from home requires a sense of trust, open communication, and a willingness to adapt to the needs of the team members. Their goal is to foster educators' self-motivation, which is useful not only for themselves but also for their colleagues and pupils at the institution. However, fewer research studies and articles relating the physical working environment to WFH policies have been published so far.

In conclusion, whether working in a traditional office or from home, the quality of the working environment can have a significant impact on employee morale and output. People who WFH anticipate a high-quality working environment with amenities such as sufficient equipment, privacy, and adequate lighting. Research indicates that a physical work environment supports an individual's work style, which has a positive effect on their telework performance. In 2020, a study discovered that the physical working environment has a significant positive effect on the performance of remote workers [71]. Consequently, it is hypothesised that a positive physical work environment has a positive effect on employee performance in a home-based work environment.

### Working from home and IT sector in Sri Lanka

During a lockdown, the IT sector is one of the most adaptable sectors to the WFH model. With the bulk of IT workers WFH, this new work practice has been supporting both local and foreign customers with continued services and even preventing economic downfall, in addition to keeping businesses afloat during lockdowns imposed from time to time. According to forecasts, the IT industry will not return to the former paradigm of office-based labour and around 75% of the workforce will continue to WFH in the nearest future. Recognising how diverse elements influence employee performance in such an environment is critical since certain factors when taken individually, have a favourable or unfavourable impact on employee performance. This study's specific objective is to gather information to understand better how communication, employment flexibility, work life balance, and the physical work environment affect employee performance in the WFH practice.

Information technology (IT) is an industry that can thrive with remote work if appropriately managed. Several academics have shown an interest in the WFH initiatives in the wake of the COVID-19 pandemic. However, the research pays little attention to how WFH impacts IT employees [72]. Therefore, there is an unfilled research vacuum in identifying the relationship between the WFH and the performance of IT Professionals in Sri Lanka. Specifically, a clear gap exists concerning the availability of literature on the factors that affect WFH in the context as explained above.

### Conceptual framework

The conceptual framework depicted in Fig 2 is designed to comprehend the study's context and the relationship between the independent variables, namely work environment, work life balance, flexible work hours, effective communication, and the dependent variable, i.e. employee performance while WFH. Teleworking has been shown in previous studies to have a favourable effect on worker performance, having resulted in higher organisational performance. The following hypotheses were derived from the conceptual framework.

**Hypothesis 1**: Effective communication is linked with a positive relationship on employee performance in a home working environment.

**Hypothesis 2**: Flexible work arrangement is linked with a positive relationship on employee performance in a home working environment.

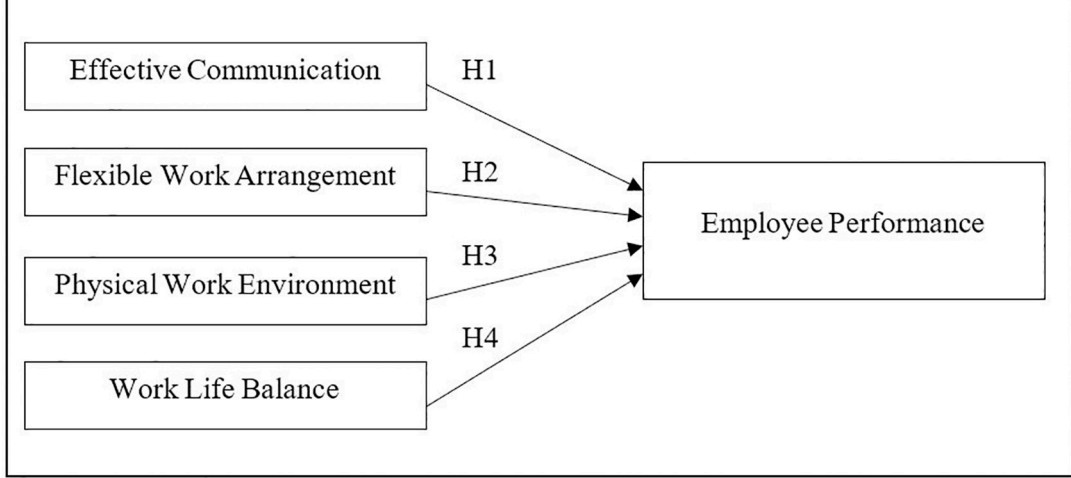

**Fig 2. Conceptual framework.** Source: Authors' illustrations.

**Hypothesis 3:** Physical work environment is linked with a positive relationship on employee performance in a home working environment.

**Hypothesis 4:** Work life balance is linked with a positive relationship on employee performance in a home working environment.

## Data and methodology

This section offers a comprehensive overview of the data, the statistical models and descriptive statistics utilised in this study.

### Data

To identify the factors affecting the performance of WFH, this research engaged HR personnel from several reputable IT organizations in Sri Lanka and solicited their opinions. This consultation was further supported by a literature review that examined previous studies related to the topic. The questionnaire, which served as the primary source of data for eliciting the perspectives of participants, was categorised into two sections. Section A focused on capturing employee demographics, whereas Section B focused on capturing the data necessary for the study's primary variables using Likert scale questions. The questionnaire is included in S1 Appendix of the supplementary materials. The Likert scale-based questions designed based on the findings of the literature review were pilot-tested with a sample of 30 participants, thereby eliminating two questions and modifying others to have a positive output. The pilot test demonstrated adequate reliability, with a Cronbach Alpha value of 0.721, leading to the distribution of the modified questionnaire to peers and networks, as depicted in Fig 3.

This study's population comprises of information technology professionals from the 173 software firms listed in the Exporter's Directory (Software Development Companies in Sri Lanka—EDB, n.d.). To achieve a 90% confidence level and a 10% error rate, a subset of 50 organisations were required from the overall population. A minimum sample size of 50 respondents is necessary for a framework with four latent variables [73]. All information technology experts employed in Sri Lanka were invited to participate in the study which consisted

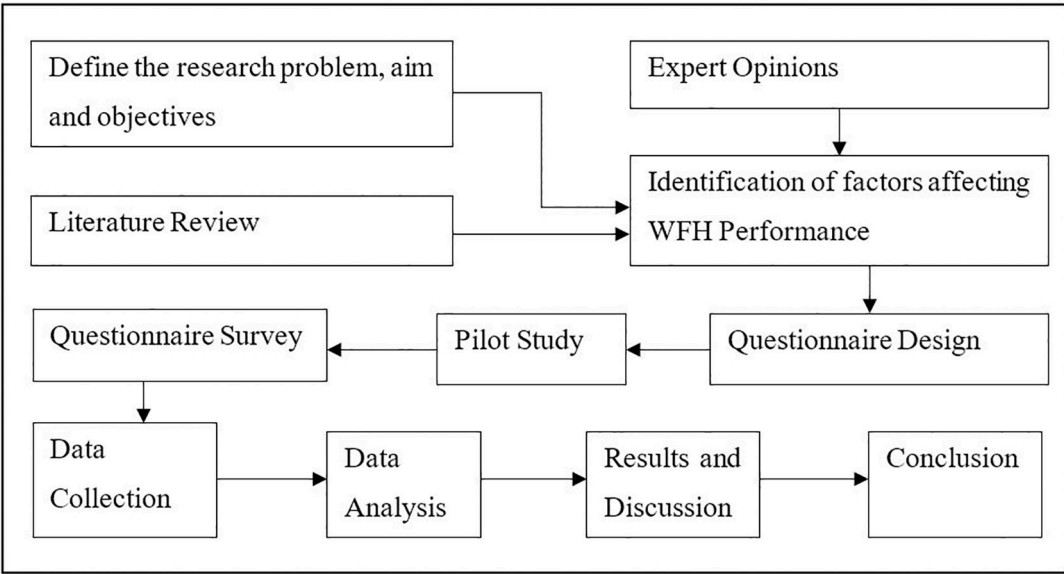

**Fig 3. Flowchart of the research process.** Source: Authors' illustrations.

**Table 1. Operationalization of the variables.**

|  | Variable Name | Indicators | Questions | References |
|---|---|---|---|---|
| Dependent Variable | Performance | Minimal supervision Quality of work Productivity | Q27, Q28, Q29, Q30 | [9, 74] |
| Independent Variables | Communication | Accessibility Effectiveness Frequency Quality of communication | Q9, Q10, Q11, Q12, Q13, Q14 | [2, 11, 28, 38] |
|  | Work Flexibility | Time Management Work Stress Mental health Degree of flexibility | Q15, Q16, Q17, Q18 | [2, 54, 55, 66, 75, 76] |
|  | Work Life Balance | Time spent on work Time spent on family Social interactions Stress | Q19, Q20, Q21, Q22 | [8, 42, 55, 77] |
|  | Physical Work Environment | Distractions Tools External Factors | Q23, Q24, Q25, Q26 | [2, 55, 65, 71] |

Source: Authors' calculations based on questionnaire created and literature

of 58 questions, and 321 responded. The questionnaire was administrated between April 2022 and October 2022, distributed through the companies social media accounts and personal contacts, utilising the internet as a means to reach IT professionals from the selected organisations. This distribution method facilitated the collection of primary data. After removing irrelevant information from the questionnaire-collected data, the study received 293 responses, supassing the minimum criteria. The data file can be obtained by referring to the S2 Appendix file under supporting documents. Table 1 describes the indicators and the questions related to each variable observed in this study.

In order to conduct quantitative research, variables must be operationalized, which entails defining how the components of interest will be quantified. This procedure requires the identification of independent and dependent variables. Table 1 provides a clear breakdown of the variables, their names, and the measurement methods for each indicator to facilitate this process. As the primary method of data collection, the Likert scale was utilized, and questionnaire items were developed based on relevant indicators derived from the reference column. The variables were effectively operationalized and reliable data were collected for analysis using this method.

## Methodology

Given that the aim of this study is to assess the likelihood of falling into either the high performance or low performance category while WFH, based on the identified independent factors of flexible work, work environment, work life balance, and effective communication, the probit model is the appropriate model to use, with the dependent variable taking the value 1 if WFH performance is high and 0 otherwise. The probit model is an estimation technique for equations with dummy dependent variables that avoids the unboundedness problem of the linear probability model by using a variant of the cumulative normal distribution. The model's form is as follows [78]:

$$P_i = \frac{1}{\sqrt{2\pi}} \int_{-\infty}^{z_i} e^{-\frac{s^2}{2dt}} \tag{1}$$

$P_i$ = probability that the dummy variable $D_i = 1$

$$Z_i = \Phi^{-1}(P_i) = \beta_0 + \beta_1 X_{1i} + \beta_2 X_{2i} + \ldots + \beta_n X_{ni} \tag{2}$$

$s$ = a standardised normal variable

Where, $\Phi-1$ is the inverse of the normal cumulative distribution function. The probit model is typically estimated by applying maximum likelihood techniques to the model, as

**Table 2. Model explanatory variables.**

| Variable | Description | Expected Sign(s) |
|---|---|---|
| *Socio-Demographic Characteristics* | | |
| gender | 1 if the respondent is Male; 0 if the respondent is Female | (+/-) |
| graduated | 1 if the respondent's level of education is graduate, postgraduate, doctoral, or above; 0 otherwise | (+/-) |
| married | 1 if the respondent is married; 0 otherwise | (-) |
| child | 1 if the respondent has one or more children; 0 otherwise | (-) |
| spouse_job | 1 if the respondent's spouse is employed; 0 otherwise | (-) |
| *Employer Demographic Characteristics* | | |
| large_company | 1 if respondent's organisation has more than 200 employees; 0 otherwise | (-) |
| flex_working_support | 1 if the respondent's organisation allows flexible working; 0 otherwise | (+/-) |
| *Working from Home Factors* | | |
| work_env | 1 if the mean of good work environment score of the respondent in WFH is greater than 3 out of 5; 0 otherwise | (+) |
| work_life_balance | 1 if the mean work life balance score of the respondent in WFH is greater than 3 out of 5; 0 otherwise | (+) |
| flex_work | 1 if the mean flexible work support score of the respondent in WFH is greater than 3 out of 5; 0 otherwise | (+/-) |
| effective_comm | 1 if the mean of effective communication score of the respondent in WFH is greater than 3 out of 5; 0 otherwise | (+) |

Source: Authors' calculations on derived variables

shown in Eq (1), and the results are presented in Eq (2). The dependent and independent variables in this study's probit model are derived as dummy variables with a value of 1 if positive and 0 if negative. These variables were extracted by computing the mean of the questions relating to those variables that were submitted on a Likert scale ranging from 1 to 5 points. Then, the dummy variables were derived by determining if the value was larger than 3 and assigning 1 value if so, and a 0 value if not. The model explanatory variables are depicted in Table 2.

The variables extracted from the collected data pertained to how the factors influence employee performance when WFH, as previously described. In addition to these WFH related variables, we also identified and derived socio-demographic characteristics from the dataset. These variables were hypothesised to impact performance based on prior research. It was expected that being married, having an employed spouse, and having children might all have a negative impact on performance. On the other hand, being a graduate identifying as and male or female were expected to have either positive or negative effects on performance, but there was no correlation between these two variables. The organisation's size and its support for flexible work were recorded as part of the organisation's demographics. Flexible work support was anticipated to have either a positive or negative impact on performance, depending on whether the organisation is an enterprise with more than 200 employees. It was also anticipated that flexible work support would positively or negatively impact performance if the organisation had more than 200 employees. Table 3 displays the dependability of the dependent and independent variables in the questionnaire.

Reliability is a scientific investigation that assesses the stability and repeatability of measurements or the capacity of a test to provide identical results under identical conditions. A reliable research instrument test consistently gives the predicted results [79]. Cronbach's alpha used to determine the instrument's reliability in the present investigation derived the total dependability score of 0.764, which is above 0.6, is acceptable for reliability [80]. As observed in Table 3, all item total correlations for the variables were greater than 0.4, indicating that the distribution of all variables was positively skewed. The work environment and communication questions appear significant, as removing these would have resulted in a reliability score of less than 0.7. The descriptive statistics of demographic variables and the dummy variables defined from the questionnaire can be observed in Tables 4 and 5 respectively.

**Table 3. Reliability analysis.**

| Variable | Corrected Item-Total Correlation | Cronbach's Alpha if Item Deleted |
|---|---|---|
| Performance | 0.4488 | 0.7503 |
| Communication | 0.5814 | 0.6929 |
| FlexibleWork | 0.4767 | 0.7360 |
| WorkLife | 0.5660 | 0.7042 |
| WorkEnvironment | 0.5813 | 0.6958 |
| Total Cronbach's Alpha = 0.7642 | | |

Source: Authors' calculations based on survey data.

Most respondents were men, indicating that the results may be skewed toward the male experience of WFH and performance in the research. In addition, a substantial proportion of the participants were married, and the majority of their spouses held employment. The majority of participants were college graduates and held engineering positions, with a significant proportion holding leadership positions within their respective organisations. Most participants were from enterprise organisations, while a smaller proportion were from Small Medium Enterprises (SMEs) and startups. Observations indicated that some organisations continue to use a hybrid model that combines office and remote work, while the majority support flexible work. The dependent and independent variables were derived using dummy variables, with 1 representing positive responses and 0 representing negative ones. The results are summarised in Table 5.

According to the derived data, the ratio of employees with high job performance to those with low job performance is close to 50/50. According to the survey, more than 85% of participants appear to have effective communication and flexible work hours. Seventy-five percent or more of the participants have a favourable work environment and flexible hours. Overall, this summary indicates that although performance is 50/50, more than 75% of independent variable values have a positive value of 1.

## Results

First, we estimated the results using the probit model as described in the methodology section. The estimation data included 293 Sri Lankan IT industry professionals from 50 companies actively involved in WFH. The results of the complete probit model estimation are presented in Table 6. The final estimation obtained through a stepwise estimation of variables are presented in Table 7. Fig 4 illustrates that the area under the receiver operating characteristic curve (ROC) is 0.6427, indicating that the estimated probit model fits well in explaining the relationship between WFH factors (such as Flexible Work, Work Environment, Work Life Balance, and Effective communication) and employee performance. The initial probit model estimation results are displayed in Table 6.

According to the initial results obtained by analysing our dataset presented in Table 6, we find that the impact of socio-demographic variables on performance is inconsistent with the indications presented in Table 2. As expected, having a spouse with a job and having children have a negative effect on employee performance, while being married has a positive effect. According to employer demographics, the size of the company has a negative effect on performance, i.e. the dependent variable or the expected outcome. With a significance level of 5%, flexible work support has also had a negative impact, contrary to the expected positive signs in the previous section. All independent variables identified as WFH factors have a positive effect on performance, supporting the hypotheses of this study. Interestingly, only work

**Table 4. Descriptive statistics of demographic variables.**

| Variable | Categories | Analytical Sample (%) | |
|---|---|---|---|
| *Employee demographics* | | | |
| Age | <25 years | 25.60 | (N = 293) |
| | 26 & 30 years | 47.44 | |
| | 31 & 40 years | 23.21 | |
| | >41 years | 03.75 | |
| Education | Diploma | 1.71 | (N = 293) |
| | Undergraduate | 7.85 | |
| | Graduate | 69.97 | |
| | Postgraduate | 20.48 | |
| Gender | Female | 18.77 | (N = 293) |
| | Male | 81.23 | |
| Civil Status | Married | 34.47 | (N = 293) |
| | Single | 65.53 | |
| Spouse job status | Employed | 65.63 | (N = 123) |
| | Unemployed | 34.38 | |
| Has children | Yes | 22.28 | (N = 193) |
| | No | 77.72 | |
| Designation | Intern | 2.39 | (N = 293) |
| | Engineer | 47.78 | |
| | Senior Engineer | 27.65 | |
| | Technical Lead | 22.28 | |
| | Architect and above | 7.51 | |
| *Employer demographics* | | | (N = 293) |
| Organization size | <10 employees | 2.73 | |
| | 10–50 employees | 16.72 | |
| | 51–200 employees | 16.04 | |
| | >200 employees | 64.51 | |
| Work from home status | Work from Home Only | 78.50 | (N = 293) |
| | Hybrid | 21.50 | |
| Support flexible work | Yes | 87.15 | (N = 293) |
| | No | 12.85 | |

Source: Authors' calculations using data collected from questionnaire.

environment and work life balance have significance on the results at a significance level of 1%, while flexible work does not. In contrast, the company's flexible working support variable has achieved statistical significance.

Next, a stepwise probit model estimation was performed to eliminate the insignificant variables and recalculate the significances and new variables were selected with p-values <0.10 and previously selected variables for removal with p-values ≥0.15. Table 7 displays the statistical output of this operation, concluding the data analysis of this study.

The coefficient estimates (Table 7) indicate that a good work environment, work life balance, and effective communication have a positive effect on the performance of a remote worker. However, it has been observed that flexible work hour support has a negative impact on employee performance although companies (i.e. employers) support them during WFH. The marginal effects demonstrate that for every 1% increase in flexible work hours support, the likelihood of decreasing the employee performance is by approximately 0.17 percentage

**Table 5. Descriptive statistics of the key variables.**

| Variable | Analytical Sample (N = 293) |
| --- | :---: |
| *Dependent variable* | |
| *Job performance* | |
| High | 46.76% |
| Low | 53.24% |
| *Independent variables* | |
| *Work environment* | |
| Good | 81.57% |
| Bad | 18.43% |
| *Work life balance* | |
| Good | 76.79% |
| Bad | 23.21% |
| *Flexible work hours* | |
| Supported | 86.01% |
| Unsupported | 13.99% |
| *Communication* | |
| Effective | 89.76% |
| Not Effective | 10.24% |

Source: Authors' calculations based on derived dummy variables.

points. Although effective communication appears on the results of the stepwise probit estimation, it is not considered significant in the final model compared to other factors. Effective communication has a positive effect on employee performance, and the marginal effect indicates that for every 1% increase in effective communication, performance rises by approximately 0.16 percentage points, which is slightly less than flexible work hour support. According to the probit model, a positive work environment and a healthy work life balance appear to be of high significance. Intriguingly, the variable, flexible work was deemed less significant, and the variable indicating flexible work hour support from the organisation achieved a significance level of less than 5%, as shown in Table 7. The coefficient estimates for both work environment and work life balance indicate that WFH performance would increase by 0.21 percentage points for every 1% increase in either work environment or work life balance.

Analysis of the ROC is the standard method for determining the sensitivity and specificity of diagnostic procedures. In our study, the area under the ROC curve is shown in both initial and final models; for brevity, Fig 4 depicts only the final model.

The 45-degree line depicts how a model without variables balances sensitivity and 1-specificity (sensitivity). The ROC curve is derived from the final model using variables. Any point on this line indicates the probability of accurately identifying pairs with high performance vs the probability of accurately predicting pairs with low performance. For instance, if sensitivity = 0.75, the likelihood of reliably recognising a pair with a high performance is 0.75, then specificity = 0.77, which represents the probability of accurately predicting a pair having low performance is 0.77. Specificity is 0.77 when sensitivity equals 0.75 and 1-specificity equals 0.23. In this instance, the area under the ROC curve is 0.6427, which shows the most recent model best reflects the independent factors explored in this study affecting the performance of WFH workers.

**Table 6. Initial probit model estimation results.**

| Variable | Estimate | Robust SE | Marginal Effect (In Percentages) |
|---|---|---|---|
| *Socio-Demographic Characteristics* | | | |
| gender | -0.01342 | 0.1948 | -0.0153 |
| graduated | 0.1346 | 0.2780 | 0.0530 |
| married | 0.2518 | 0.3225 | 0.1000 |
| child | -0.1334 | 0.2617 | -0.0526 |
| spouse_job | -0.3199 | 0.3027 | -0.1253 |
| *Employer Demographic Characteristics* | | | |
| large_company | -0.0328 | 0.1623 | -0.0130 |
| flex_working_support | -0.4931** | 0.2410 | -0.1941** |
| *Working from Home Factors* | | | |
| work_env | 0.5288** | 0.2144 | 0.2014*** |
| work_life_balance | 0.5417*** | 0.1941 | 0.2073*** |
| flex_work | 0.1323 | 0.2488 | 0.0521 |
| effective_comm | 0.4715* | 0.2914 | 0.1790* |
| Log likelihood | -188.1562 | | |
| Number of Observations | 293 | | |
| Pseudo R$^2$ | 0.0707 | | |

Note

*** significant at the 1% level

** significant at the 5% level

* significant at 10% level.

## Discussion

The study results lead to the conclusion that the physical work environment plays the most crucial role. It significantly enhances job performance when individuals are WFH. Moreover, the suitability of the work environment at home is the most important element determining the performance of remote work [81]. This is further supported by the study's findings, which disclosed that the working environment has a favourable and statistically significant impact on employee performance [82]. This suggests that in a WFH setting, the work environment can

**Table 7. Probit model stepwise estimation results.**

| Variable | Estimate | Robust SE | Marginal Effect (In Percentages) |
|---|---|---|---|
| *Employer Demographic Characteristics* | | | |
| flex_working_support | -0.4484** | 0.2235 | -0.1770** |
| *Working from Home Factors* | | | |
| work_env | 0.5645*** | 0.2104 | 0 .2141*** |
| work_life_balance | 0.5491*** | 0.1911 | 0.2101*** |
| effective_comm | 0.4223 | 0.2829 | 0.1615 |
| Log likelihood | -189.1218 | | |
| Number of Observations | 293 | | |
| Pseudo R$^2$ | 0.066 | | |

Note

*** significant at the 1% level

** significant at the 5% level.

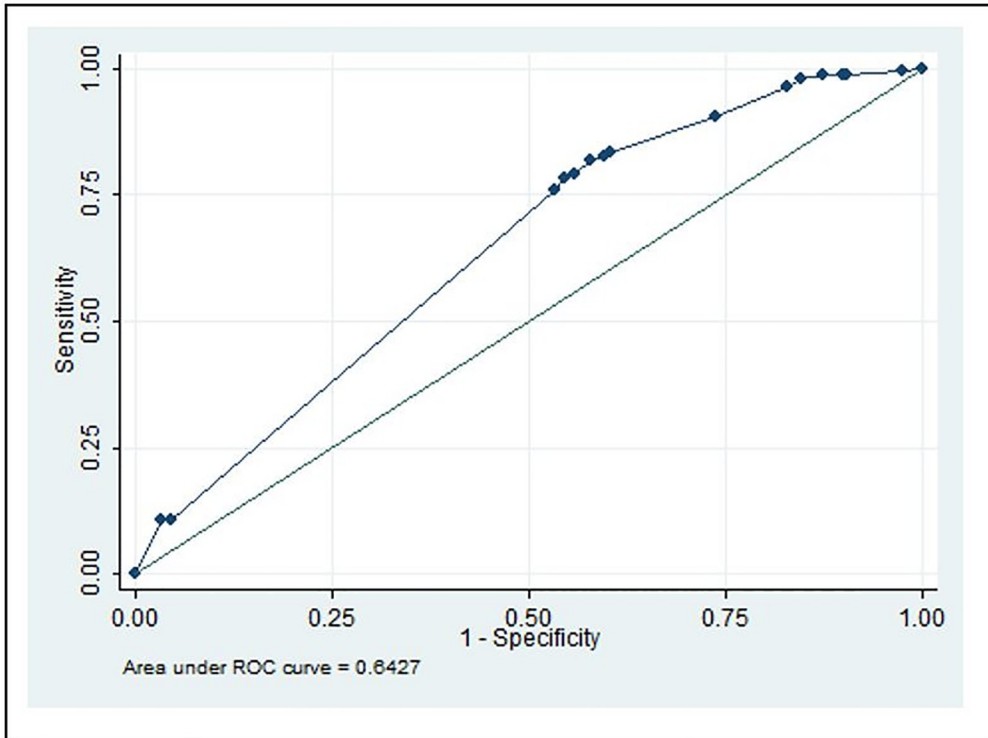

**Fig 4. ROC curve of the final model.** Source: Authors' illustrations.

enhance job performance, thus proving Hypothesis 3 in this study. Generally, the IT sector focuses on finding digital solutions to a vast range of organisational IT services, from operational issues to sustaining competitive advantage–on data processing, supply chain, logistics, among others. Engineers are the key employees in IT sector firms responsible for performing such tasks; therefore, they must be at ease in terms of mindset and the appropriate work atmosphere to do so. Moreover, IT is not a sector where employees are given a limited number of repetitive duties unlike in other sectors, like retail, labour, or manufacturing. Instead, the nature of job role and responsibilities require IT specialists who handle these to have an innovative approach to solving complex issues. In other words, the results of this study complement those of earlier studies and, more particularly, the IT sector.

Furthermore, the results of this study demonstrate that work life balance significantly affects how well remote IT employees perform, confirming Hypothesis 4 of the study. Working from home considerably and negatively impacts work life balance which lowers job satisfaction, as highlighted in an Indonesian-based study. The study further suggested that setting boundaries between work and personal life at home was challenging, particularly in a pandemic setting with several limitations [27]. A study reveals that telecommuting in the digital workplace may enhance employees' creativity if effective work life balance strategies are devised and implemented by employers [83]. However, performance in the IT industry is mainly assessed based on task completion, i.e., how many tasks are completed on time. Whether a work life balance exists or not, engineers ensure that assignments are completed as scheduled. Many engineers have integrated their work into their daily lives, and their close associates have developed a tolerance for them. A work life balance is desirable for engineers since it boosts their energy and lowers their stress levels. The nature of IT jobs demands that engineers fulfil the tasks they are committed to within the allocated tight timeframes, even

without a work life balance, making it irrelevant to their performance in a real-life work setting. However, by reducing stress and enhancing the engineer's sense of well-being, work life balance may help accomplish tasks more efficiently.

The effectiveness of virtual communication, according to the study's findings, is the component with the least impact on work output when WFH. This could be because employees typically work in teams at their places of employment, and employee performance is influenced by the ability of the team and individual co-workers to WFH. Collaboration becomes problematic when staff members cannot exchange essential work-related information [84]. Individual employees who WFH would have to put in more effort to share their knowledge when co-workers are not nearby. This result highlights the need to consider how co-workers' physical presence and collaboration affect one another and how they utilise one another's knowledge and skills, as these aspects affect performance at the individual level [85]. These results confirm Hypothesis 1 and the prior literature also explains the significance of virtual communication for remote workers' job effectiveness. Employee effectiveness in the IT industry, as well as in any other sector, depends on virtual communication. Software development teams, in particular, need to continually plan and develop features, architectures, and cooperate to produce high-quality software. While software developers may prefer working along, this is not the case for larger projects, where engineers must collaborate and complete their specific tasks to generate ideas, develop solutions, and meet project deadlines. All of these aspects depend on the level of communication between team members, and better communication leads to better employee performance. In other words, the results of this study align with previous research, particularly in the IT sector.

The results of this study indicate that work flexibility does not significantly affect job performance when WFH, contradicting Hypothesis 2 of the study. Teleworkers have more choice in how, when, and under what circumstances they can perform tasks better because they are not physically being watched [60], which increases employee flexibility over job expectations [22]. It has previously been established that a positive mental attitude is crucial to the accomplishments of a software developer. The ability to adapt one's work environment to changing circumstances helps maintain mental steadiness. Engineers want to operate without the supervision of others while still collaborating with others, and flexible work support enables all of this. However, the observation that flexible work support negatively impacts performance is mainly because in large organisations, IT professionals work in teams and collaborate. Flexible work may help provide peace of mind for individual workers, but as a team, performance may be hindered the more there are blockers and dependencies on other team members. Therefore, for a team involved with a common deliverable, flexible work support may negatively affect their performance as a unit. These explanations indicate that this study's results complement those of earlier studies, particularly those focused on the IT sector.

This study's significance extends beyond the boundaries of traditional research on remote work and job performance. While existing literature has primarily focused on developed economies and their well-established information technology sectors, this research delves into the intricacies of WFH within the context of a developing nation. Sri Lanka's burgeoning information and communications technology (ICT) sector, though promising, faces unique challenges and opportunities. As developing countries like Sri Lanka make strides in the IT industry, they encounter distinct scenarios that demand an in-depth investigation. The IT professionals operating in these regions often navigate a complex landscape, where innovation and resourcefulness are essential to address the diverse demands of clients from across the globe. The practicality and adaptability of remote work, as explored in this study, emerge as vital determinants of their effectiveness. Furthermore, the study underscores the specific nuances of IT work in a developing country, where adaptability, self-reliance, and creative problem-solving

skills are paramount. Unlike standardized, repetitive tasks found in certain sectors, the IT domain demands adaptability and innovation to tackle multifaceted issues. The results of this research spotlight how these qualities, often intrinsic to IT professionals in such settings, interact with the components of remote work.

By illuminating these interactions and their impact on job performance, this study offers not only a deeper understanding of remote work but also practical insights for emerging economies. As the global workforce evolves, bridging the gap between established practices and novel challenges is critical. The significance of this research lies in its potential to inform strategies for managing remote work in developing countries, thereby contributing to the robustness and adaptability of their IT sectors. The findings provide a valuable resource for policymakers, organizations, and IT professionals alike as they navigate the ever-evolving landscape of remote work.

## Conclusion

The results of this study which focused on IT specialists in the IT business, reveal that improved communication, a conducive work environment and work life balance are factors that enhance employee performance. These findings confirm Hypotheses 1,3 and 4 in the study and support IT professionals in succeeding when WFH. It is also revealed that flexible work hours may negatively affect performance, indicating that Hypothesis 2 is not supported due to the target-driven, stressful nature of IT work that requires high concentration. Workplace performance, role stress, and perceived autonomy benefit from telecommuting. Most earlier studies projected the notion that WFH is elective and voluntary, offering better freedom in choosing the place for work. In the absence of non-voluntary WFH situations, increased stress from work-family problems is a common adverse outcome. A timely shift is predicted for WFH because the IT sector can swiftly adapt to a WFH situation. This is supplemented by the fact that most employees have the necessary skills and knowledge to embrace and perform better with new technologies that facilitate remote working.

This study prioritised increasing the sample size and heavily relied on quantitative data to meet the research objectives within the allotted timeframe. However, this approach may have impeded the identification of a more nuanced comprehension of the diverse outcomes associated with remote work performance. Therefore, future research should adopt mixed-method and qualitative approaches to gain a more complete understanding of the variables influencing remote work performance. In addition, the analysis in this study was limited to a set of factors including effective communication, work life balance, a comfortable workplace, and flexible work assistance. However, to improve the validity of remote work policies, future research should investigate a broad range of variables that influence the efficiency of remote IT personnel. A more comprehensive approach that enables organisations to optimise their remote work policies and foster better work experiences for their employees, would be advantageous for both the employees and the organisation as a whole.

### Limitations

The study was conducted in 2022, during the ongoing COVID-19 outbreak. The uncertainty brought on by the pandemic has seen people experiencing ample problems, triggering anxiety about whether they can continue living and maintaining their health. As a direct consequence of this, individuals did not respond to the surveys with much enthusiasm. The lockdown in place at the time of the study also proved to be a barrier in interacting with a greater percentage of the populace. The sample size of the study was confined to Sri Lanka. Despite this limitation, the sample size was considered sufficient as it represented the population of IT professionals in

Sri Lanka. However, for a more comprehensive understanding of the impact of working from home policies in the current era, future research should encompass other countries. Nonetheless, the findings of this study offer valuable insights into the experiences of individuals in Sri Lanka during the pandemic.

While this study provides valuable insights into the dynamics of WFH among IT professionals in Sri Lanka, there are several promising avenues for future research to further enrich our understanding of remote work in the ever-evolving landscape. Firstly, exploring cross-cultural variations in remote work experiences can unveil how cultural nuances shape the effectiveness of WFH policies, leading to more tailored approaches for different regions. Secondly, conducting multinomial analysis of employee performance, particularly by breaking down results based on distinct job designations, can provide a more nuanced understanding of how remote work impacts IT professionals at different career stages and roles. Thirdly, an examination of the specific stressors, coping mechanisms, and the impact on mental health among IT professionals working from home could offer critical insights for enhancing remote work policies. Additionally, studying the adoption of emerging digital tools and platforms in the remote work setting can provide insights into the changing technological landscape of the industry. Lastly, a comparative analysis between pandemic and post-pandemic remote work environments can reveal enduring adaptations and evolving trends in the realm of remote work. These research directions hold the potential to inform more adaptable and culturally sensitive remote work policies and practices for IT professionals and organizations in the future.

## Policy recommendations

Having a high-functioning remote working team in IT is possible because of IT-enabled online platforms' support and IT professionals' expertise. Using such platforms facilitates the sharing of knowledge and information to an extent where employees interact more and/or get involved with and feel less isolated, communication is the most influential factor impacting performance in a WFH situation. Such an environment aids in performance optimisation. Considering the foregoing, the authors suggest that employers pay attention to the following:

- Employers must provide improved communication infrastructure to lessen communication hurdles like interrupted internet connectivity, lagging video, voice disturbances, etc.

- Consider the potential for implementing new procedures to enhance and maintain ongoing connection and communication between remote workers and their managers, team leaders, and employers.

- Offer financial support to maintain a better working environment, including computer hardware, provide electronic devices and high-speed internet connections, and train employees to conduct virtual meetings and brainstorming for complex problem-solving and timely task completion.

- Review the organisation's current rules, procedures, and practices to identify out dated ones and replace these by implementing flexible ones for more employee participation. Examples include interactive online classes and seminars, online training courses, and events like virtual staff get together activities.

- Review the existing performance evaluation and awarding methods to address employees' challenges while exhibiting their success when WFH. As suggested by literature and findings, a flexible work environment and poor communication can result in negative impacts. To minimise negative consequences of a full-time WFH approach, a hybrid model in the office for physical meetings and verbal communication can be effective in the long run.

## Supporting information

**S1 Appendix. Questionnaire.**
(DOCX)

**S2 Appendix. Data file.**
(XLSX)

## Acknowledgments

The authors would like to thank Ms. Gayendri Karunarathne for proof-reading and editing this manuscript.

## Author Contributions

**Conceptualization:** Nilesh Jayanandana, Ruwan Jayathilaka.

**Data curation:** Nilesh Jayanandana.

**Formal analysis:** Nilesh Jayanandana, Ruwan Jayathilaka.

**Methodology:** Nilesh Jayanandana, Ruwan Jayathilaka.

**Project administration:** Ruwan Jayathilaka.

**Software:** Nilesh Jayanandana.

**Supervision:** Ruwan Jayathilaka.

**Validation:** Nilesh Jayanandana, Ruwan Jayathilaka.

**Visualization:** Nilesh Jayanandana.

**Writing – original draft:** Nilesh Jayanandana, Ruwan Jayathilaka.

**Writing – review & editing:** Ruwan Jayathilaka.

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
