## [Decision Letter · Decision Letter 0]

27 Mar 2023

PONE-D-22-32645Working from Home Reshaping the Work Practices: Factors Affecting Job Performance of Sri Lankan IT Professionals Working from HomePLOS ONE

Dear Dr. Jayathilaka,

Thank you for submitting your manuscript to PLOS ONE. After careful consideration, we feel that it has merit but does not fully meet PLOS ONE’s publication criteria as it currently stands. Therefore, we invite you to submit a revised version of the manuscript that addresses the points raised during the review process.

We look forward to receiving your revised manuscript.

Kind regards,

Katarzyna Piwowar-Sulej

Academic Editor

PLOS ONE

Journal Requirements:

2. In the ethics statement in the Methods, you have specified that verbal consent was obtained. Please provide additional details regarding how this consent was documented and witnessed, and state whether this was approved by the IRB

Reviewers' comments:

Reviewer's Responses to Questions

**Comments to the Author**

1. Is the manuscript technically sound, and do the data support the conclusions?

Reviewer #1: Yes

Reviewer #2: Partly

2. Has the statistical analysis been performed appropriately and rigorously? 

Reviewer #1: Yes

Reviewer #2: Yes

3. Have the authors made all data underlying the findings in their manuscript fully available?

Reviewer #1: Yes

Reviewer #2: Yes

4. Is the manuscript presented in an intelligible fashion and written in standard English?

Reviewer #1: Yes

Reviewer #2: Yes

5. Review Comments to the Author

Reviewer #1: Dear Authors,

I am not sure if your references and their citation in the manuscript follow the journal template or expectations. Your work needs minor modifications. Please consider citation of the following works: https://link.springer.com/chapter/10.1007/978-3-030-40417-8_9 and https://doi.org/10.1016/j.procs.2022.09.354

The factual figures and correctly performed inference and calculations are to be commended.

Reviewer #2: Thank you for the opportunity to read the manuscript entitled “Working from Home Reshaping the Work Practices: Factors Affecting Job Performance of Sri Lankan IT Professionals Working from Home”. The research concerns a valid and important topic.

However, I believe that there are some major issues with the current form of the manuscript that limit its potential. Below I outline the some concerns I had while reading the manuscript in the hope that they can help in further development:

1. The title needs rewording. The first part of the title suggests that the research is about how WFH changes work (that the research will have two states before WFH and after its introduction). What is not true.

2. The abstract also needs rewriting. The beginning of the abstract is misleading as it suggests that the authors are addressing the issue of how WFH affects job performance. From the remainder of the abstract, it appears that selected factors affecting work-from-home performance are being investigated.

3. In the abstract, please include also 2-3 special quantitative achievements from the findings of this study.

4. The “Introduction” section needs a few more sentences to strengthen the article. Please add theoretical framework - indication of the specific theory on which the article is based.

5. Please include the research problem, objective and questions in the last paragraph of the “Introduction”.

6. The description of the systematic literature review lacks details, e.g. the timeframe in which the articles were published, the languages, whether only peer-reviewed articles were considered, or if there were any further narrowing down.

7. The results presented in Figure 1 are not described in the body of the article. Please describe them.

8. Please revise the literature review section. It is advisable that the paragraph titles correspond to the variables in the model. Please complete the section on the relationship of WFH to the physical work environment

9. Hypotheses should follow directly from the literature. Please move them directly under the relevant sections with a description of the results of the literature review.

10. Please separate the section on the specificity of WFH in the IT sector.

11. Please provide detailed stages of the research process.

12. Please add a table with the structure of the research sample.

13. Please add information about the research tools used, in particular where the questions came from (who formulated them and by whom they were validated beforehand) and on which scales they were answered.

14. The inclusion of the model equation is not necessary. I suggest deletion.

15. Please complete the description of the variables, e.g. what questions were included.

16. Please separate the “Results” section and please explain research problems, solutions, and the theoretical contribution of your study in the section.

17. In the section with “Conclusions” add paragraphs mentioning the limitations of the study and remedies to limitations.

18. Please add to the “Conclusions” section the future scope of your research.

19. Please correct typos in the text.

6. PLOS authors have the option to publish the peer review history of their article (what does this mean?). If published, this will include your full peer review and any attached files.

Reviewer #1: No

Reviewer #2: No

---

## [Author Response · Author response to Decision Letter 0]

10 Apr 2023

Point by point response to editor and reviewers

Dear editor and the reviewers,

We would like to express our profound appreciation to the editor and the reviewers for the valuable comments and suggestions made on our manuscript which were very helpful in revising and improving it.

Please note that the line numbers referred in this document is aligned with the revised manuscript which has track changes.

Reviewer 1 comment 1: I am not sure if your references and their citation in the manuscript follow the journal template or expectations. Your work needs minor modifications. Please consider citation of the following works: https://link.springer.com/chapter/10.1007/978-3-030-40417-8_9 and https://doi.org/10.1016/j.procs.2022.09.354

The factual figures and correctly performed inference and calculations are to be commended.

Authors’ Response to Reviewer 1 comment 1: Thank you very much for the comment. PLOS ONE uses “Vancouver” style” for the references. Authors have revised the references to match the PLOS ONE format as below link. 

https://journals.plos.org/plosone/s/submission-guidelines#loc-reference-style

Reviewer 2 comment 1: The title needs rewording. The first part of the title suggests that the research is about how WFH changes work (that the research will have two states before WFH and after its introduction). What is not true.

Authors’ Response to Reviewer 2 comment 1: Thank you very much. Well noted and revised the title to “Factors Affecting Job Performance of Sri Lankan IT Professionals Working from Home” – Line 25 to 27.

Reviewer 2 comment 2: The abstract also needs rewriting. The beginning of the abstract is misleading as it suggests that the authors are addressing the issue of how WFH affects job performance. From the remainder of the abstract, it appears that selected factors affecting work-from-home performance are being investigated

Authors’ Response to Reviewer 2 comment 2: Thank you very much for your feedback. Authors have rewritten the abstract to better describe the summary of the paper. This change is available in line 29 to 42.

“This study investigated the influence of physical work environment, work-life balance, work flexibility, and effective communication on the job performance of IT professionals in Sri Lanka's IT industry who work from home (WFH). A standard questionnaire was used to collect data from 293 IT specialists in 50 different IT organizations in Sri Lanka, and a stepwise probit model was used for data analysis. According to the findings, the physical work environment and work-life balance had a significant positive effect on job performance. A one-unit increase in physical work environment and work-life balance increased the likelihood of high job performance by 0.21% and 0.19%, respectively. In contrast, work flexibility had a negative effect on job performance, with an increase of one unit resulting in a 0.18% decrease in the likelihood of high job performance. The positive impact of effective communication on job performance was less significant. The study emphasises the significance of providing a conducive work environment and promoting work-life balance to improve the job performance of IT professionals in Sri Lanka's IT industry who WFH”

Reviewer 2 comment 3: In the abstract, please include also 2-3 special quantitative achievements from the findings of this study.

Authors’ Response to Reviewer 2 comment 3: Thank you very much for your comment and authors have added quantitative achievements into the abstract to better summarize the paper.

This change is available in line 35 to line 39.

“…A one-unit increase in physical work environment and work-life balance increased the likelihood of high job performance by 0.21% and 0.19%, respectively. In contrast, work flexibility had a negative effect on job performance, with an increase of one unit resulting in a 0.18% decrease in the likelihood of high job performance. The positive impact of effective communication on job performance was less significant...”

Reviewer 2 comment 4: The “Introduction” section needs a few more sentences to strengthen the article. Please add theoretical framework - indication of the specific theory on which the article is based

Authors’ Response to Reviewer 2 comment 4: Duly noted and thank you very much. Authors have added the following paragraph to describe the theoretical framework of the study in the introduction section. This change is available in line 150 to 162.

“The theoretical framework for the study is based on the job demands-resources (JD-R) model, which suggests that job demands can lead to job strain and negative outcomes, while job resources can lead to job engagement and positive outcomes [14]. In this study, WFH is a job demand, whereas effective communication, work flexibility, work-life balance, and physical work environment are job resources. The study hypothesises that these resources can mitigate the adverse effects of WFH and boost job performance. Specifically, effective communication can reduce isolation and improve collaboration, physical work environment can provide a comfortable and conducive workspace, work flexibility can provide autonomy and control, and work-life balance can decrease work-family conflict and improve overall well-being. The JD-R model provides a theoretical lens through which the complex interaction between job demands and job resources, as well as how these can impact job performance in the context of WFH for IT professionals can be comprehended in Sri Lanka's IT industry.”

Reviewer 2 comment 5: Please include the research problem, objective and questions in the last paragraph of the “Introduction”.

Authors’ Response to Reviewer 2 comment 5: Thank you very much for your input. Authors have added the following paragraph to address this comment. This change is available in line 110 to 131.

“The COVID-19 pandemic has significantly altered how organisations operate, with remote work gaining popularity. Further, WFH policies have been implemented in worldwide to ensure business continuity and employee safety. Southeast Asian nation like Sri Lanka is not an exception to this trend. The country's thriving information and communications technology (ICT) sector heavily depends on remote work practices. However, little attention has been paid to how WFH affects the performance of IT professionals in Sri Lanka. Despite the abundant research on remote work, a focus on the performance of IT professionals during WFH is necessary, particularly in Sri Lanka. Filling this void, the present study focuses on the effects of effective communication, flexible work schedules, work-life balance, and physical work environment on the performance of employees during WFH. This study will be guided by the following research questions: How does effective communication affect employee performance during WFH?; How does a flexible work schedule impact WFH employee performance?; How does work-life balance impact the performance of employees during WFH?; and how does the physical work environment influence WFH employee performance?. By addressing these research questions, this study will contribute to gain a better understanding of the factors influencing the performance of IT professionals during WFH. Thereby, the study assists organisations and policymakers in Sri Lanka in strengthening their policy management and practices regarding remote work. This study also refers research work by multiple scholars during a 22-year timespan in 1998-2020 [4, 9-13].”

Reviewer 2 comment 6: The description of the systematic literature review lacks details, e.g. the timeframe in which the articles were published, the languages, whether only peer-reviewed articles were considered, or if there were any further narrowing down.

Authors’ Response to Reviewer 2 comment 6: Thank you very much for your kind observation. The authors have updated Figure 1 (box two) with the following content.

" …limited to January 1985 December 2021 (inclusive) and Peer Reviewed and English Language...”

Reviewer 2 comment 7: The results presented in Figure 1 are not described in the body of the article. Please describe them.

Authors’ Response to Reviewer 2 comment 7: Duly Noted and Thank you very much. Authors have updated the literature review section taking your inputs into consideration. Following is the added paragraph. This change is available in line 182 to 191.

“A total of 126 online publications with full text written in English language and peer reviewed were identified through the period from January 1985 to December 2021. The abstracts were then reviewed and discarded if: (i) these were irrelevant to the topic (n=21), (ii) these lacked sufficient information (n=15), or (iii) these were duplicates of previously published articles (n=35). This procedure led to the exclusion of 35 publications. Data were extracted from the remaining 55 publications and categorised into five subject areas: work-life balance (n=10), work flexibility (n=7), effective communication (n=12), physical work environment (n=8), and performance (n=18). To conclude the methodology, 55 empirical case studies were selected under the aforementioned five areas, and their key characteristics are summarised below.”

Reviewer 2 comment 8: Please revise the literature review section. It is advisable that the paragraph titles correspond to the variables in the model. Please complete the section on the relationship of WFH to the physical work environment

Authors’ Response to Reviewer 2 comment 8: Thank you very much for your kind observation. We have reviewed and expanded literature review to strengthen the relationship between work from home and physical work environment. This change is available in line 382 to 449.

“Working from Home and Physical Work Environment

When looking at the many elements that influence WFH, it was discovered that decreased interpersonal interactions, decreased management support and trust, and the appropriateness of the work atmosphere at home-office were highly important [46]. Workers who WFH can personalise their environment to meet their own specifications, requirements and preferences, which can improve job performance [47]. A study investigating the effect of WFH on job performance during the COVID-19 crisis in Indonesia stated that WFH allows workers to create a customised work environment tailored to their preferences and way of life [48]. Employee performance in WFH scenarios is positively influenced by a physically acceptable work environment, which has a substantial and favourable effect on performance. Having a well-maintained working environment with appropriate space, a peaceful ambience, good lighting, and improved working equipment enable employees achieve even greater levels of productivity [14, 49-51]. Additional studies indicated that workplace factors and operational commitment had an impact on job performance, both directly and indirectly when WFH, which support earlier results [52-54].

The Journal of Occupational and Environmental Medicine published a cross-sectional study on the performance of home-based workers in relation to their actual work environment in 2021. The study was conducted in Japan on workers who spent at least one day per month working from home. When working with WFH, the physical work space was a source of exposure. The Work Functioning Impairment Scale was used to assess the existence of work-related impairment. A "No" answer to suggested surroundings was shown to be a significant predictor of work functional impairment. Answering "No" to the question and "Is there enough light to perform my job?" were related to the greatest odds ratio of work functional impairment [28].

A paper published in 2020 disclosed that teachers' self-motivation in a remote work environment is influenced by various factors. These include their perceptions of the setting's ability to meet their psychological requirements for autonomy, competence, and interpersonal connection. To assist team leaders in being more responsive to the needs of their subordinates, the study have provided a list of practical tips [55] . Working from home requires a sense of trust, open communication, and a willingness to adapt to the needs of the team members. Their goal is to foster educators' self-motivation, which is useful not only for themselves but also for their colleagues and pupils at the institution. However, fewer research studies and articles relating the physical working environment to WFH policies have been published so far.

In conclusion, whether working in a traditional office or from home, the quality of the working environment can have a significant impact on employee morale and output. People who WFH anticipate a high-quality working environment with amenities such as sufficient equipment, privacy, and adequate lighting. Research indicates that a physical work environment supports an individual's work style, which has a positive effect on their telework performance. In 2020, a study discovered that the physical working environment has a significant positive effect on the performance of remote workers [56]. Consequently, it is hypothesised that a positive physical work environment has a positive effect on employee performance in a home-based work environment.

Reviewer 2 comment 9: Hypotheses should follow directly from the literature. Please move them directly under the relevant sections with a description of the results of the literature review.

Authors’ Response to Reviewer 2 comment 9: Thank you for your kind comment. Authors have revised the literature review and have added following for each sub section of literature review. These changes are available in line 289 to 294, line 317 to 325, line 375 to 381 and line 439 to 449.

Working from Home and Effective Communication

“…In summary, effective communication is essential for home-based workers, as it helps combat feelings of social isolation and disengagement. Clear communication regarding work expectations, job responsibilities, goals, objectives, and deadlines increases job satisfaction, loyalty, and productivity. Virtual communication is rife with miscommunication, which can result in less productivity during WFH. Consequently, it is hypothesised that effective communication has a positive effect on employee performance in a home-based office..”

Working from Home and Flexible Work

“…In conclusion, flexible work arrangements, such as WFH or working outside of normal business hours, are associated with a positive work attitude and high job performance. These arrangements assist workers in maintaining a healthy work-life balance, thereby enhancing job satisfaction and productivity. There is a correlation between flexible work arrangements and job performance, according to previous studies. The purpose of these arrangements is to assist employees in striking a balance between their personal and professional lives (i.e. work-life balance), which can lead to increased productivity. Therefore, it is hypothesised that flexible work arrangements improve employee performance in a home-based office.”

Working from Home and Work Life Balance

“…In summary behind the idea of "work-life balance" is that an individual's personal and professional lives should complement one another. It has been discovered that remote work has both positive and negative effects on work-life balance, which can affect worker performance and productivity. According to prior research, a lack of assistance in balancing professional and personal responsibilities can increase stress levels and result in subpar performance. Therefore, it is hypothesised that a healthy work-life balance has a positive effect on the performance of employees who WFH..”

Working from Home and Physical Work Environment

“…In conclusion, whether working in a traditional office or from home, the quality of the working environment can have a significant impact on employee morale and output. People who WFH anticipate a high-quality working environment with amenities such as sufficient equipment, privacy, and adequate lighting. Research indicates that a physical work environment supports an individual's work style, which has a positive effect on their telework performance. In 2020, a study discovered that the physical working environment has a significant positive effect on the performance of remote workers [56]. Consequently, it is hypothesised that a positive physical work environment has a positive effect on employee performance in a home-based work environment.”

Reviewer 2 comment 10: Please separate the section on the specificity of WFH in the IT sector.

Authors’ Response to Reviewer 2 comment 10: Duly Noted! We have added a section in the literature review called “Working from Home and IT Sector in Sri Lanka” to address this issue. This change is available from line 450 onwards.

Reviewer 2 comment 11: Please provide detailed stages of the research process.

Authors’ Response to Reviewer 2 comment 11: Thank you for your feedback. Authors have added “Figure 3: Flowchart of the Research Process” to explain the research process. This change is available in line 514.

Reviewer 2 comment 12: Please add a table with the structure of the research sample.

Authors’ Response to Reviewer 2 comment 12: Thank you very much for your feedback. Authors have added Table 4: Descriptive Statistics of Demographic Variables to explain the structure of the research sample. Furthermore, the following paragraph was added to explain the table. This change is available in line 608 to 620.

“Most respondents were men, indicating that the results may be skewed toward the male experience of WFH and performance in the research. In addition, a substantial proportion of the participants were married, and the majority of their spouses held employment. The majority of participants were college graduates and held engineering positions, with a significant proportion holding leadership positions within their respective organisations. Most participants were from enterprise organisations, while a smaller proportion were from Small Medium Enterprises (SMEs) and startups. Observations indicated that some organisations continue to use a hybrid model that combines office and remote work, while the majority support flexible work. The dependent and independent variables were derived using dummy variables, with 1 representing positive responses and 0 representing negative ones. The results are summarised in Table 5.”

Reviewer 2 comment 13: Please add information about the research tools used, in particular where the questions came from (who formulated them and by whom they were validated beforehand) and on which scales they were answered.

Authors’ Response to Reviewer 2 comment 13: Thank you very much for the feedback. Authors have addressed this with following paragraph. This change is available in line 498 to 511.

“To identify the factors that affect the performance of WFH, this research consulted the HR personnel from several well-known IT organizations in Sri Lanka and sought their opinion. This consultation was further supported by a literature review that examined previous studies related to the topic. The questionnaire, which served as the primary source of data for eliciting the perspectives of participants, was categorised into two sections. Section A focused on capturing employee demographics, whereas Section B focused on capturing the data necessary for the study's primary variables using Likert scale questions. The questionnaire is included in Appendix S1 of the supplementary materials. The Likert scale-based questions designed based on the findings of the literature review were pilot-tested with a sample of 30 participants, thereby eliminating two questions and modifying others to have a positive output. The pilot test demonstrated adequate reliability with a Cronbach Alpha value of 0.721, resulting in the distribution of the modified questionnaire to peers and networks as depicted in Figure 3.”

Reviewer 2 comment 14: The inclusion of the model equation is not necessary. I suggest deletion.

Authors’ Response to Reviewer 2 comment 14: Thank you for the comments. Authors used equation, since it can convey complex ideas and help to make scientific writing more precise and rigorous. By using equations, authors can clearly define the variables, parameters, and relationships involved in their work, which can help to ensure that their results are accurate and reproducible. Thus, authors would like to keep the model equation in the revised manuscript.

Reviewer 2 comment 15: Please complete the description of the variables, e.g. what questions were included.

Authors’ Response to Reviewer 2 comment 15: Thank you for your feedback. Well noted and we have added a table under data section with the title “Table 1: Operationalization of the Variables”. This change is available in line 540.

Reviewer 2 comment 16: Please separate the “Results” section and please explain research problems, solutions, and the theoretical contribution of your study in the section.

Authors’ Response to Reviewer 2 comment 16: Duly noted and thank you very much for your comment. Authors have separated Results and Discussion section in the revised manuscript address this issue. Results section is now available from line 628 to 697. Discussion section is available from line 699 to 772. These are two separate sessions in the revised manuscript.

Reviewer 2 comment 17: In the section with “Conclusions” add paragraphs mentioning the limitations of the study and remedies to limitations.

Authors’ Response to Reviewer 2 comment 17: Thank you for your feedback and we have added a section to address limitations to the revised manuscript. This change is available in line 804 to 811.

“Limitations

The study was conducted in 2022, at a time when the COVID-19 outbreak was ongoing. The uncertainty brought on by the pandemic has seen people suffering ample problems, triggering anxiety on whether people can continue living and upkeep their health. As a direct consequence of this, individuals did not answer to the surveys with much enthusiasm. The lockdown being in place at the time of the study has also proven to be a barrier in interacting with a greater percentage of the populace.”

Reviewer 2 comment 18: Please add to the “Conclusions” section the future scope of your research.

Authors’ Response to Reviewer 2 comment 18: Duly noted and we have added a paragraph to conclusion adding the future scope of the research. This change is available in line 791-803.

“This study prioritised increasing the sample size and heavily relied on quantitative data to meet the research objectives within the allotted timeframe. Nonetheless, this approach may have impeded in identifying a more nuanced comprehension of the diverse outcomes associated with remote work performance. Therefore, future research should adopt a mixed-method and qualitative approaches to gain a more complete understanding of the variables influencing remote work performance. In addition, the analysis of this study was limited to a set of factors including effective communication, work-life balance, a comfortable workplace, and flexible work assistance. However, to improve the validity of remote work policies, future research should investigate a broad range of variables that influence the efficiency of remote IT personnel. A more comprehensive approach that enables organisations to optimise their remote work policies and foster better work experiences for their employees, would be advantageous for both the employees and the organisation as a whole.”

Reviewer 2 comment 19: Please correct typos in the text.

Authors’ Response to Reviewer 2 comment 19: Thank you very much for your kind remarks and authors have rechecked and done a proofread on the revised manuscript.

---

## [Decision Letter · Decision Letter 1]

30 May 2023

PONE-D-22-32645R1Factors Affecting Job Performance of Sri Lankan IT Professionals Working from HomePLOS ONE

Dear Dr. Jayathilaka,

Thank you for submitting your manuscript to PLOS ONE. After careful consideration, we feel that it does not fully meet PLOS ONE’s publication criteria as it currently stands. Therefore, we invite you to submit a revised version of the manuscript that addresses the points raised during the review process and additionally - adresses the Editor's concerns. The Editor can not promise that the final decision will be positive.

We look forward to receiving your revised manuscript.

Kind regards,

Katarzyna Piwowar-Sulej

Academic Editor

PLOS ONE

Editor's and Reviewers' comments:

Editor: This paper lacks strong theoretical argument. When searching in databases, the authors missed many relevant papers (related, e.g., to telecommuting, virtual work, work from home, lockdown, Covid-19). Many articles have been published on the topic (job performance, telework) which referred to the Covid-19 pandemic. So, the research gap is not enough justified. You neither present the papers that undertook similar problems in your literature review nor compare their findings with your findings. Therefore, In my opinion, your contribution to science is not significant. Furthermore, the article which you used are not up-do-date. Please, enrich your paper by using relevant, high quality literature sources.

Reviewer #1: The study under review presents original research, but its rationale for conducting such research is not clear. The paper fails to provide convincing reasons to undertake this study, leaving the reader wondering about its significance. Additionally, it appears that some parts of the research have already been published in a previous study by the same authors, and the similarity between the two papers is too high. The reported results have already been made public, which raises questions about the originality and novelty of the study under review.

Despite these shortcomings, the study's experiments, statistics, and other analyses are of high technical quality. However, the paper's technical descriptions are inadequate and do not provide sufficient detail to replicate the experiments or verify the results. The references list is also a cause for concern as it has not been revised or extended. The absence of an updated references list casts doubt on the paper's scholarly rigor, and it may indicate that the authors did not conduct a thorough literature review.

Reviewer #2:

1. Please correct some typos, e.g. in the keywords and section title there is "work life balance" (without hyphen) in the text "work-life balance" (with hyphen). I suggest alignment.

2. I did not notice in the body of the article when exactly (month and year) the questionnaire survey was conducted. Please add the infomation.

3. Using the term 'impact' in hypotheses is usually risky. I would suggest replacing it with 'linked by a positive relationship'.

4. I would suggest expanding the limitations section, e.g. in terms of the limitations of research methodology used.

---

## [Author Response · Author response to Decision Letter 1]

13 Jul 2023

Point by point response to editor and reviewers

Dear editor and the reviewers,

We would like to express our profound appreciation to the editor and the reviewers for the valuable comments and suggestions made on our manuscript which were very helpful in revising and improving it.

Please note that the line numbers referred in this document is aligned with the revised manuscript which has track changes.

Editor’s comments: This paper lacks strong theoretical argument. When searching in databases, the authors missed many relevant papers (related, e.g., to telecommuting, virtual work, work from home, lockdown, Covid-19). Many articles have been published on the topic (job performance, telework) which referred to the Covid-19 pandemic. So, the research gap is not enough justified. You neither present the papers that undertook similar problems in your literature review nor compare their findings with your findings. Therefore, in my opinion, your contribution to science is not significant. Furthermore, the article which you used are not up-do-date. Please, enrich your paper by using relevant, high quality literature sources.

Authors’ Response to Editor’s comment: Thank you very much for your comment. This research was conducted in 2021/2022. We have added new references to the latest papers, including those from 2022 and 2023, in the most recent submission. We have also reviewed papers that reference the following query

Search Query: ("work life balance” or “virtual office” or “virtual platform” or “work from home” or “WFH” or “work life” or “telework’’) and ("communication” or “performance” or “efficiency” or “flexibility” or “balance” or “IT” or “Information Technology”)) limited to January 1985 March 2023 (inclusive) and Peer Reviewed and English Language.

As a research gap, we have identified the insufficient amount of research conducted on IT professionals in Sri Lanka regarding the effects of working from home on performance. We believe that the culture and mindset of people vary from country to country, emphasizing the need for conducting this research within the specific context of Sri Lanka.

In the original submission, we had 55 references, which we subsequently increased to 72 in the first revision. In the second revision, we further incorporated additional studies and increased the reference count to 87.

Reviewer 1 comment 1: The study under review presents original research, but its rationale for conducting such research is not clear. The paper fails to provide convincing reasons to undertake this study, leaving the reader wondering about its significance. Additionally, it appears that some parts of the research have already been published in a previous study by the same authors, and the similarity between the two papers is too high. The reported results have already been made public, which raises questions about the originality and novelty of the study under review.

Despite these shortcomings, the study's experiments, statistics, and other analyses are of high technical quality. However, the paper's technical descriptions are inadequate and do not provide sufficient detail to replicate the experiments or verify the results. The references list is also a cause for concern as it has not been revised or extended. The absence of an updated references list casts doubts on the paper's scholarly rigor, and it may indicate that the authors did not conduct a thorough literature review.

Authors’ Response to Reviewer 1 comment 1: Thank you for bringing to our attention the issue with the reference count not being updated in the submitted paper. We sincerely apologize for the oversight in not updating the reference number in Figure 1 during the previous iteration. The original submission indeed had 55 references, and the first revision included 72 references (17 new references). However, regrettably, the authors failed to update the reference count in Figure 1 for the first revision. In the second revision, we have included a total of 87 references (15 new references) and have rectified the reference count in Figure 1 accordingly. 

The following references were added in the second revision.

29. Tooranloo, Azadi, & Sayyahpoor, 2017

30. Anakpo, Nqwayibana, & Mishi, 2023

31. Ahmad, Asmawi, & Samsi, 2022

40. Abdullah, Rahmat, Zawawi, Khamsah, & Anuarsham, 2020

41. Hamad et al. 2021

42. Kleine, Rudolph, & Zacher, 2019

47. Sousa-Uva et al., 2021

48. Grincevičienė, 2020

54. Pollitt, 2008

55. Raišiene, Rapuano, Varkulevičiute, & Stachová, 2020

57. Abioro, Oladejo, & Ashogbon, 2018

56 Golden, TDVeiga, & Dino, 2008

50. Ammons & Markham, 2004

1. Baker et al., 2006

59. Holtz & Harold, 2008

After conducting an extensive literature review, we have identified a research gap in the existing literature regarding the effects of working from home (WFH) on the job performance of IT professionals in Sri Lanka. In light of this gap, our research focuses exclusively on this particular group of professionals to investigate the impact of WFH on their performance. To the best of our knowledge, no previous studies have specifically addressed this research question within the context of Sri Lanka.

We would like to emphasize that our research differs from the study you mentioned in several key aspects. These distinctions encompass the sample coverage and data collection period, as well as the respondents and data analysis techniques employed. As a result, we believe our study makes a unique contribution to the field by addressing this specific research gap and providing fresh insights into the performance of IT professionals in Sri Lanka within the WFH context.

We hope that this clarification adequately addresses your question. If you have any further concerns or require additional information, please feel free to ask. We appreciate your positive comments on the analysis. Thank you.

Reviewer 2 comment 1: Please correct some typos, e.g. in the keywords and section title there is "work life balance" (without hyphen) in the text "work-life balance" (with hyphen). I suggest alignment

Authors’ Response to Reviewer 2 comment 1: Thank you very much. Well noted and revised the entire article to have “work life balance” 

Reviewer 2 comment 2: I did not notice in the body of the article when exactly (month and year) the questionnaire survey was conducted. Please add the information.

Authors’ Response to Reviewer 2 comment 2: Thank you for your feedback. The authors have taken your suggestion into account and have now included the time period during which the survey was conducted. This change can be found in lines 395 to 398 of the revised paper with track changes.

“The questionnaire was conducted between April 2022 to October 2022. It was distributed through the company's social media accounts and personal contacts, utilising the internet as a means to reach IT professionals from the selected organisations. This distribution method allowed for the collection of primary data.”

Reviewer 2 comment 3: Using the term 'impact' in hypotheses is usually risky. I would suggest replacing it with 'linked by a positive relationship'.

Authors’ Response to Reviewer 2 comment 3: Thank you for your comment. The authors have made changes to the wording of the hypothesis based on your suggestions. This modification can be found in lines 478 to 485 of the revised paper with track changes.

“Hypothesis 1: Effective communication is linked with a positive relationship on employee performance in a home working environment. 

Hypothesis 2: Flexible work arrangement is linked with a positive relationship on employee performance in a home working environment. 

Hypothesis 3: Physical work environment is linked with a positive relationship on employee performance in a home working environment. 

Hypothesis 4: Work life balance is linked with a positive relationship employee performance in a home working environment.”

Reviewer 2 comment 4: I would suggest expanding the limitations section, e.g. in terms of the limitations of research methodology used

Authors’ Response to Reviewer 2 comment 4: Thank you for your comment. The authors have included the following information in the limitation section, specifically in lines 778 to 785 of the revised paper with track changes.

“The study was conducted in 2022, at a time when the COVID-19 outbreak was ongoing. The uncertainty brought on by the pandemic has seen people suffering ample problems, triggering anxiety on whether people can continue living and upkeep their health. As a direct consequence of this, individuals did not answer to the surveys with much enthusiasm. The lockdown being in place at the time of the study has also proven to be a barrier in interacting with a greater percentage of the populace. The sample size of the study confined to Sri Lanka. Despite this limitation, the sample size was considered sufficient as it represented the population of IT professionals in Sri Lanka. However, for a more comprehensive understanding of the impact of working from home policies in the current era, future research should encompass other countries. Nonetheless, the findings of this study offer valuable insights into the experiences individuals in Sri Lanka during the pandemic.

---

## [Decision Letter · Decision Letter 2]

20 Oct 2023

PONE-D-22-32645R2Factors Affecting Job Performance of Sri Lankan IT Professionals Working from HomePLOS ONE

Dear Dr. Jayathilaka,

Thank you for submitting your manuscript to PLOS ONE. After careful consideration, we feel that it has merit but does not fully meet PLOS ONE’s publication criteria as it currently stands. Therefore, we invite you to submit a revised version of the manuscript that addresses the points raised during the review process.

We look forward to receiving your revised manuscript.

Kind regards,

Katarzyna Piwowar-Sulej

Academic Editor

PLOS ONE

Journal Requirements:

Reviewers' comments:

Reviewer's Responses to Questions

**Comments to the Author**

1. If the authors have adequately addressed your comments raised in a previous round of review and you feel that this manuscript is now acceptable for publication, you may indicate that here to bypass the “Comments to the Author” section, enter your conflict of interest statement in the “Confidential to Editor” section, and submit your "Accept" recommendation.

Reviewer #2: All comments have been addressed

Reviewer #3: (No Response)

2. Is the manuscript technically sound, and do the data support the conclusions?

Reviewer #2: Yes

Reviewer #3: Yes

3. Has the statistical analysis been performed appropriately and rigorously? 

Reviewer #2: Yes

Reviewer #3: Yes

4. Have the authors made all data underlying the findings in their manuscript fully available?

Reviewer #2: Yes

Reviewer #3: Yes

5. Is the manuscript presented in an intelligible fashion and written in standard English?

Reviewer #2: Yes

Reviewer #3: Yes

6. Review Comments to the Author

Reviewer #2: Thank you for making the corrections. The following suggestions are primarily the result of the evolving nature of research on the topics under discussion and the emergence of new findings:

1. In response to the reviews, the authors have emphasized the contextual significance focusing on the Sri Lankan workforce. Therefore, it is advisable to incorporate "Sri Lanka" as a keyword while eliminating the redundant terms "performance" and "employee performance."

2. The dynamics surrounding remote work are subject to rapid changes, with e.g. several global corporations presently advocating for a return to office-based work. Consequently, the introductory section may require an update or clarification to reflect the ongoing trend of expanding remote work in Sri Lanka.

3. In the Introduction, it would be valuable to elucidate the necessity of conducting a distinct study on a Sri Lankan sample. Specifically, articulating the distinctive features of the Sri Lankan workforce that remain unexplored in previous studies conducted elsewhere would enhance the article's context.

4. Recent investigations on productivity in remote work environments have revealed a decline in performance. For instance, studies indicate that employees assigned to full-time remote work exhibit an 18% decrease in productivity compared to their office counterparts. It is advisable to enrich the article with these contemporary findings. For additional information, refer to the study available at https://www.nber.org/system/files/working_papers/w31515/w31515.pdf.

5. The section dedicated to "Working from Home and Flexible Work" encompasses various subtopics, including work-life balance (which is the subject of the subsequent section). The central theme of work flexibility may not be immediately evident. It is recommended to rephrase this section to ensure that the core concept of work flexibility is prominently highlighted.

Reviewer #3: In the introduction section it is written "The theoretical framework for the study is based on the job demands-resources (JD-R)". Can this be clarified in more depth so that it does not become ambiguous, or can it also be expanded in the literature section.

Of all the existing findings, the discussion section does not clearly explain the novelty and value of the contribution (Need a special paragraph).

For future research there are no suggestions regarding methodological aspects.

7. PLOS authors have the option to publish the peer review history of their article (what does this mean?). If published, this will include your full peer review and any attached files.

Reviewer #2: No

Reviewer #3: **Yes: **Yandra Rivaldo

---

## [Author Response · Author response to Decision Letter 2]

29 Oct 2023

Point by point response to editor and reviewers

Dear editor and the reviewers,

We would like to express our profound appreciation to the editor and the reviewers for the valuable comments and suggestions made on our manuscript which were very helpful in revising and improving it.

Please note that the line numbers referred in this document is aligned with the revised manuscript which has track changes.

Reviewer 2 comment 1: In response to the reviews, the authors have emphasized the contextual significance focusing on the Sri Lankan workforce. Therefore, it is advisable to incorporate "Sri Lanka" as a keyword while eliminating the redundant terms "performance" and "employee performance."

Authors’ Response to Reviewer 2 comment 1: Thank you for your valuable feedback. We have removed the keyword 'employee performance' and have added the keywords 'Sri Lanka' and 'Information Technology' to line 42.

Reviewer 2 comment 2: The dynamics surrounding remote work are subject to rapid changes, with e.g. several global corporations presently advocating for a return to office-based work. Consequently, the introductory section may require an update or clarification to reflect the ongoing trend of expanding remote work in Sri Lanka.

Authors’ Response to Reviewer 2 comment 2: Thank you for your thoughtful observation. This study was conducted in 2022 when the global trend favored remote work. However, as of 2023, there has been a shift towards office-based work. We have updated the introduction section in lines 100-103 to reflect this change:

“…It is worth noting that in 2023, organizations have started to shift back to office work policies, and WFH is not encouraged as much as it used to be. This study aims to provide better insight into that decision and explore what factors have affected work-from-home for IT professionals in Sri Lanka..…”

Reviewer 2 comment 3: In the Introduction, it would be valuable to elucidate the necessity of conducting a distinct study on a Sri Lankan sample. Specifically, articulating the distinctive features of the Sri Lankan workforce that remain unexplored in previous studies conducted elsewhere would enhance the article's context.

Authors’ Response to Reviewer 2 comment 3: Thank you for your valuable suggestion. We have incorporated a new paragraph in the introduction section (lines 106-118) to provide context for the study:

“In focusing on the Sri Lankan context, it is essential to highlight the unique circumstances and emerging status of the country in the realm of information technology. Sri Lanka, as a developing nation, presents a dynamic landscape where the IT sector is swiftly evolving. It is pertinent to recognize that Sri Lanka's IT literacy and workforce, in comparison to other developing countries, are factors that have remained unexplored in previous studies conducted elsewhere. For instance, exploring how Sri Lanka ranks in terms of IT literacy and its burgeoning IT sector compared to other developing countries can shed light on the distinctive features of the Sri Lankan workforce. According to the World Bank, Sri Lanka's IT sector has grown by 20% annually in recent years, outpacing the global average of 10%. The country's IT literacy rate is 60%, which is higher than other developing countries in the region, such as India (40%) and Bangladesh (25%) [14]. These factors, in conjunction with the global shift towards WFH, underscore the need for a focused examination of the performance of IT professionals in Sri Lanka during remote work scenarios.”

Reviewer 2 comment 4: Recent investigations on productivity in remote work environments have revealed a decline in performance. For instance, studies indicate that employees assigned to full-time remote work exhibit an 18% decrease in productivity compared to their office counterparts. It is advisable to enrich the article with these contemporary findings. For additional information, refer to the study available at https://www.nber.org/system/files/working_papers/w31515/w31515.pdf.

Authors’ Response to Reviewer 2 comment 4: Thank you for this valuable recommendation. We have enriched the literature review in lines 250-267 with the following additions:

“In a study done in 2017 with a sample size of 235 data entry workers in India, the authors explored how working from home affected worker productivity. They found that workers who were randomly assigned to work from home were 18% less productive than those who worked in the office. This difference was mostly due to the fact that workers who worked from home learned more slowly. The authors also found that workers who preferred to work from home were more productive than those who preferred to work in the office. However, this difference was not enough to offset the negative effects of working from home. Overall, the paper finds that working from home can have a negative impact on worker productivity. However, the authors also noted that there are some benefits to working from home, such as improved work-life balance and reduced commuting time [35]. In contrast, A study by the Sri Lanka Association of Software and Services Companies (SLASSCOM) found that Sri Lankan IT professionals performed well during remote work scenarios. The study found that 90% of IT professionals were able to maintain or improve their productivity while working remotely. Additionally, 85% of IT professionals reported that they were satisfied with their work-life balance while working remotely. Overall, the Sri Lankan IT sector is a dynamic and growing sector with a young and adaptable workforce [36]. IT professionals in Sri Lanka have performed well during remote work scenarios, demonstrating the country's potential to become a major player in the global IT industry.”

Reviewer 2 comment 5: The section dedicated to "Working from Home and Flexible Work" encompasses various subtopics, including work-life balance (which is the subject of the subsequent section). The central theme of work flexibility may not be immediately evident. It is recommended to rephrase this section to ensure that the core concept of work flexibility is prominently highlighted.

Authors’ Response to Reviewer 2 comment 5: Thank you for your suggestion. We have revised the section on 'Working from Home and Flexible Work' to emphasise the core concept of work flexibility. The changes are made in lines 349-382.

“Recent research illuminates the diverse dimensions of work flexibility and its implications for employees. In a study conducted in 2021, secondary school teachers in Ekiti State displayed a modest level of work-life balance, work flexibility, and job efficiency, underscoring the interconnectedness of work-related stress, workload, and job performance. Work-flexibility, in this context, significantly influenced work life balance, which in turn, influenced the teachers' job performance. Furthermore, insights gained from investigations into remote work experiences during the initial wave of the COVID-19 pandemic in Portugal underscored the impact of the work environment and flexible work culture on telework happiness [50]. Comprehensive research is required to further understand telework satisfaction and its effects on physical and mental health, facilitating the development of effective strategies for enhancing employee well-being and cultivating a conducive teleworking environment. As the scope of telework and mobile work arrangements continues to expand due to advances in digitalization, it reshapes working conditions and job quality. A study published in the International Journal of Environmental Research and Public Health delved into the impact of various telework forms on job quality. The research revealed that gender, the nature of telework, flexible hours in telework and the intensity of information and communication technology (ICT) usage significantly shape working conditions and job quality [51]. Notably, home-based teleworkers who regularly work from home reported the highest job quality, underscoring the benefits of work flexibility. However, this achievement sometimes comes at the cost of lower skill development, decision-making autonomy, pay, and career opportunities. This research aims to investigate current data on job quality and work flexibility, with a particular focus on gender as a critical aspect of analysis. Regarding employee performance, a flexible work arrangement was found to have no direct impact. 

As the demands of both professional and personal life grow, so does the need for effective time management and work overload. An individual's ability to function well at work is adversely affected by factors such as job overload and poor time management. In a study published in the Dynamic Relationships Management Journal [52], researchers looked at how time management affects job performance and the link between work overload and flexible working hours which lead to work life balance. The findings revealed that effective time management reduces the negative effects of work overload and poor job performance when combined. It means workplace productivity and work–life balance suffers because of overwork. Considering these results, it is essential for both people (employees) and companies (employers) to pay greater attention to time management to promote work–life balance and productivity.”

Reviewer 3 comment 1: In the introduction section it is written "The theoretical framework for the study is based on the job demands-resources (JD-R)". Can this be clarified in more depth so that it does not become ambiguous, or can it also be expanded in the literature section.

Authors’ Response to Reviewer 3 comment 1: Thank you for your valuable feedback. To provide a clearer understanding of the theoretical framework, we have included a paragraph in the literature review (lines 160-184) that elaborates on the Job Demands-Resources (JD-R) model.

“The theoretical underpinning of this study is grounded in the Job Demands-Resources (JD-R) model, an influential framework in the field of organizational psychology and work-related research. Developed to examine the interplay between workplace elements and employee well-being, the JD-R model posits a fundamental distinction between two categories of factors: job demands and job resources [15]. Job demands encompass the aspects of work that require sustained physical or psychological effort and are associated with the potential for strain and negative outcomes. In the context of this study, the act of WFH serves as a pertinent job demand, given its potential to introduce challenges and stressors into the professional lives of IT professionals in Sri Lanka. On the other side of this theoretical spectrum are job resources, which are the elements of work that facilitate the achievement of work goals, reduce job strain, and promote positive outcomes. In this study, we identify four critical job resources that play a pivotal role in enhancing the job performance of IT professionals during WFH: effective communication, work flexibility, work-life balance, and the physical work environment. Effective communication fosters collaboration and alleviates the isolation often associated with remote work. Work flexibility empowers employees with autonomy and control over their work schedules and conditions. Work-life balance addresses the delicate equilibrium between personal and professional life, a crucial determinant of overall well-being. The physical work environment, even in the context of remote work, provides a comfortable and conducive workspace for IT professionals. The JD-R model serves as an invaluable theoretical lens through which the complex interplay between job demands and job resources is examined. It offers a comprehensive perspective on how these factors can influence job performance in the unique setting of WFH for IT professionals within Sri Lanka's IT industry. Through this framework, this study seeks to unravel the intricate dynamics of remote work and offer insights that can guide organizations and policymakers in enhancing work conditions and performance for IT professionals in an evolving work landscape [15].”

Reviewer 3 comment 2: Of all the existing findings, the discussion section does not clearly explain the novelty and value of the contribution (Need a special paragraph).

Authors’ Response to Reviewer 3 comment 2: Thank you for your valuable feedback. In response to your suggestion, we have included the following paragraph in the discussion section (lines 751-774) to highlight the novelty and significance of this study.

“This study's significance extends beyond the boundaries of traditional research on remote work and job performance. While existing literature has primarily focused on developed economies and their well-established information technology sectors, this research probes into the intricacies of WFH within the context of a developing nation. Sri Lanka's burgeoning information and communications technology (ICT) sector, though promising, faces unique challenges and opportunities. As developing countries like Sri Lanka make strides in the IT industry, they encounter distinct scenarios that demand an in-depth investigation. The IT professionals operating in these regions often navigate a complex landscape, where innovation and resourcefulness are essential to address the diverse demands of clients from across the globe. The practicality and adaptability of remote work, as explored in this study, emerge as vital determinants of their effectiveness. Furthermore, the study underscores the specific nuances of IT work in a developing country, where adaptability, self-reliance, and creative problem-solving skills are paramount. Unlike standardized, repetitive tasks found in certain sectors, the IT domain demands adaptability and innovation to tackle multifaceted issues. The results of this research spotlight how these qualities, often intrinsic to IT professionals in such settings, interact with the components of remote work.

By illuminating these interactions and their impact on job performance, this study offers not only a deeper understanding of remote work but also practical insights for emerging economies. As the global workforce evolves, bridging the gap between established practices and novel challenges is critical. The significance of this research lies in its potential to inform strategies for managing remote work in developing countries, thereby contributing to the robustness and adaptability of their IT sectors. The findings provide a valuable resource for policymakers, organizations, and IT professionals alike as they navigate the ever-evolving landscape of remote work.”

Reviewer 3 comment 3: For future research there are no suggestions regarding methodological aspects.

Authors’ Response to Reviewer 3 comment 3: For future research there are no suggestions regarding methodological aspects. Thank you for your feedback. To provide more insight into future research, we have included Cross-Cultural Variations, Multinomial Analysis, Stressors and Mental Health, Digital Tool Adoption and Comparative Analysis. These research directions hold the potential to inform more adaptable and culturally sensitive remote work policies and practices for IT professionals and organizations in the future. This can be seen in the revised manuscript in lines 869-884:

“While this study provides valuable insights into the dynamics of WFH among IT professionals in Sri Lanka, there are several promising avenues for future research to further enrich our understanding of remote work in the ever-evolving landscape. Firstly, exploring cross-cultural variations in remote work experiences can unveil how cultural nuances shape the effectiveness of WFH policies, leading to more tailored approaches for different regions. Secondly, conducting multinomial analysis of employee performance, particularly by breaking down results based on distinct job designations, can provide a more nuanced understanding of how remote work impacts IT professionals at different career stages and roles. Thirdly, an examination of the specific stressors, coping mechanisms, and the impact on mental health among IT professionals working from home could offer critical insights for enhancing remote work policies. Additionally, studying the adoption of emerging digital tools and platforms in the remote work setting can provide insights into the changing technological landscape of the industry. Lastly, a comparative analysis between pandemic and post-pandemic remote work environments can reveal enduring adaptations and evolving trends in the realm of remote work. These research directions hold the potential to inform more adaptable and culturally sensitive remote work policies and practices for IT professionals and organizations in the future.”

---

## [Editor Report · Decision Letter 3]

21 Nov 2023

Factors Affecting Job Performance of Sri Lankan IT Professionals Working from Home

PONE-D-22-32645R3

Dear Dr. Jayathilaka,

We’re pleased to inform you that your manuscript has been judged scientifically suitable for publication and will be formally accepted for publication once it meets all outstanding technical requirements.

Kind regards,

Katarzyna Piwowar-Sulej

Academic Editor

PLOS ONE

---

## [Editor Report · Acceptance letter]

22 Nov 2023

PONE-D-22-32645R3 

Factors Affecting Job Performance of Sri Lankan IT Professionals Working from Home 

Dear Dr. Jayathilaka:

I'm pleased to inform you that your manuscript has been deemed suitable for publication in PLOS ONE. Congratulations! Your manuscript is now with our production department. 

Kind regards, 

on behalf of

Professor Katarzyna Piwowar-Sulej 

Academic Editor

PLOS ONE